# A monthly 1-degree resolution dataset of daytime cloud fraction over the Arctic during 2000–2020 based on multiple satellite products

Xinyan Liu[1], Tao He[1], Shunlin Liang[2], Ruibo Li[3], Xiongxin Xiao[4], Rui Ma[1], Yichuan Ma[1]

[1] School of Remote Sensing and Information Engineering, Wuhan University, Wuhan 430079, China

[2] Department of Geography, University of Hong Kong, Hong Kong 999077, China

[3] State Key Laboratory of Remote Sensing Science, Aerospace Information Research Institute, Chinese Academy of Sciences, Beijing 100101, China

[4] Institute of Geography and Oeschger Center for Climate Change Research, University of Bern, Bern 3012, Switzerland

*Correspondence to*: Tao He (taohers@whu.edu.cn)

**Abstract.** The low accuracy of satellite cloud fraction (CF) data over the Arctic seriously restricts the accurate assessment of the regional and global radiative energy balance under a changing climate. Previous studies have reported that no individual satellite CF product could satisfy the needs of accuracy and spatio-temporal coverage simultaneously for long-term applications over the Arctic. Merging multiple CF products with complementary properties can provide an effective way to produce a spatiotemporally complete CF data record with higher accuracy. This study proposed a spatiotemporal statistical data fusion framework based on cumulative distribution function (CDF) matching and the Bayesian maximum entropy (BME) method to produce a synthetic $1°\times1°$ CF dataset in the Arctic during 2000–2020. The CDF matching was employed to remove the systematic biases among multiple passive sensor datasets through the constraint of using CF from an active sensor. The BME method was employed to combine adjusted satellite CF products to produce a spatiotemporally complete and accurate CF product. The advantages of the presented fusing framework are that it not only uses the spatiotemporal autocorrelations but also explicitly incorporates the uncertainties of passive sensor products benchmarked with reference data, i.e., active sensor product and ground-based observations. The inconsistencies of Arctic CF between passive sensor products and the reference data were reduced by about 10–20% after fusing, with particularly noticeable improvements in the vicinity of Greenland. Compared with ground-based observations, $R^2$ increased by about 0.20–0.48, the root mean square error (RMSE) and bias reductions averaged about 6.09% and 4.04% for land regions, respectively; these metrics for ocean regions were about 0.05–0.31, 2.85%, and 3.15%, respectively. Compared with active sensor data, $R^2$ increased by nearly 0.16, and RMSE and bias declined by about 3.77% and 4.31%, respectively, in land; meanwhile, improvements in ocean regions were about 0.3 for $R^2$, 4.46% for RMSE and, 3.92% for bias. The results of the comparison with the ERA5 and the MRI-AGCM3-2-S climate model suggest an obvious improvement in the consistency between the satellite-observed CF and the reanalysis and model data after fusion. This serves as a promising indication that the fused CF results hold the potential to deliver reliable satellite observations for modeling and reanalysis data. Moreover, the fused product effectively supplements the temporal gaps of AVHRR-based products caused by satellite faults and the data missing from MODIS-based products prior to the launch of Aqua, and extends

the temporal range better than the active product; it addresses the spatial insufficiency of the active sensor
data and the AVHRR-based products acquired at latitudes greater than 82.5 °N. A continuous monthly 1-
degree CF product covering the entire Arctic during 2000–2020 was generated and is freely available to
the public at https://doi.org/10.5281/zenodo.7624605 (Liu et al., 2022). This is of great importance for
reducing the uncertainty in the estimation of surface radiation parameters and thus helps researchers to
better understand the earth's energy imbalance.
**1 Introduction**
Clouds substantially affect Earth's energy budget by reflecting solar radiation back to space and by
restricting emissions of thermal radiation into space (Ramanathan et al., 1989; Van Tricht et al., 2016;
Danso et al., 2020). Clouds are also an essential variable in the climate system because they are directly
associated with precipitation and aerosol loading (Toll et al., 2019; Poulsen et al., 2016). The cloud
fraction (CF), which represents the amount of sky estimated to be covered by a specified cloud type or
level (partial CF) or by all cloud types and levels (total CF), has long been recognized as a major source
of uncertainty when estimating radiation flux and future climate change (Xie et al., 2010; Liu et al., 2011a;
Qian et al., 2012; Danso et al., 2020). An accurate representation of CF is essential for the evaluation of
regional and global energy budgets as well as for predicting future climatic conditions. However,
variances in CF definitions and system differences commonly exist among different sources of data. As
a solution, the fused product provides a higher level of definition consistency and accuracy in comparison
to alternative datasets.
By making spatially continuous observations, satellites provided us with an unprecedented
advantage in assessing regional and global cloud effects. In the last few decades, increased effort has
been made to develop, analyze, and validate global or regional cloud property datasets that are based on
long-term satellite observations (Heidinger et al., 2014; Hollmann et al., 2013; Karlsson and Devasthale,
2018; Marchant et al., 2016; Rossow and Schiffer, 1999; Stubenrauch et al., 2013; Enriquez-Alonso et
al., 2016; Sun et al., 2015; Tzallas et al., 2019; Wu et al., 2014). Studies have also shown that although
different cloud datasets were derived from different observation instruments and algorithms, most of
them provide quite consistent CF observations in middle and lower-latitude regions (Karlsson and
Devasthale, 2018; Stengel et al., 2017; Claudia, 2012). However, systematic errors and artifacts exist in
CF data, so some inconsistencies inevitably occur among different datasets (Sun et al., 2015; Tzallas et
al., 2019; Wu et al., 2014), especially in the polar regions (Liu et al., 2022). Perennial snow/ice coverage
coupled with frequent moisture inversions in Arctic has limited the cloud detection capabilities of passive
sensor datasets, where the differences between these various datasets tend to be about two-fold in
magnitude when compared with datasets acquired at other latitudes (Karlsson and Devasthale, 2018; Liu
et al., 2022; Stubenrauch et al., 2013). The uncertainties of the annual global surface downward longwave
(LW) and shortwave (SW) fluxes caused by satellite-derived cloud properties were calculated at about
2% (7 $\mathrm{Wm^{-2}}$ and 4 $\mathrm{Wm^{-2}}$, respectively) and those for global surface upward LW and SW were about
0.8% (about 3 $\mathrm{Wm^{-2}}$) and 13% (also 3 $\mathrm{Wm^{-2}}$), respectively (Kato et al., 2011; Kato et al., 2012; Kim
and Ramanathan, 2008). It should be noted that the differences in CF may have a more obvious impact
on the surface radiation budget in high-latitude polar regions. Kennedy et al. (2012) found that the CF
bias might cause monthly biases in Arctic surface SW and LW fluxes over 90 and 60 $\mathrm{Wm^{-2}}$ for some
reanalyses, respectively (Kennedy et al., 2012). Walsh et al. (2009) proposed that the bias of summer
low-level CF would create deviations of about 160 $\mathrm{Wm^{-2}}$ in estimated SW radiation (Walsh et al., 2009).
Some other related studies have also found that the variances of annual Arctic surface radiation
estimation caused by CF uncertainty were higher than 10 $Wm^{-2}$ (Hakuba et al., 2017; Kato et al., 2018b;
Huang et al., 2017). Therefore, relying on a single CF dataset may introduce large uncertainty when
analyzing the cloud dynamics over the Arctic, further affecting the estimated energy budget and related
climate applications.
Each cloud dataset has its own advantages and disadvantages in Arctic CF detection. The Advanced
Very High Resolution Radiometer (AVHRR) offers the longest continuous satellite observation records
extending from 1978 to the present and provides daily global coverage based on data from several
AVHRRs. With the successful operation of new generations of satellites, the frequency of global view
has increased to more than eight each day, which provides richer angular information for CF observations
(Heidinger et al., 2014; Karlsson et al., 2017). Many cloud products exist that are based on AVHRR
sensors. The International Satellite Cloud Climatology Project (ISCCP) H-series product relies on newer
passive imagers with higher spectral, spatial, radiometric, and temporal resolutions; it provides revised
daytime cloud detection over snow and ice in polar regions (Young et al., 2018). Moreover, the ISCCP
is largely unaffected by the AVHRR orbital drifts (Loyola R et al., 2010; Liu et al., 2022). The CM SAF
cLoud, Albedo, and RAdiation datasets (CLARA-A1/A2) systematically use CALIPSO-CALIOP cloud
information for development and validation purposes, and it optimizes the detection conditions during
the polar day over snow- and ice-covered surfaces (Karlsson et al., 2017; Karlsson and Hakansson, 2018).
The AVHRR Pathfinder Atmospheres - Extended (PATMOS-x) product is the first multi-parameter
dataset that is making use of all AVHRR channels. This product has a relatively finer spatial resolution
than other AVHRR-based records, and it also improves cloud detection based on active sensor data
(Heidinger et al., 2012; Heidinger et al., 2014). However, the AVHRR-based products are often reported
to underestimate Arctic CF because of the limitations in radiation correction and spatial bands (Stengel
et al., 2017; Kotarba, 2015). In addition, the United States National Oceanic and Atmospheric
Administration's (NOAA's) archiving of data has its own problems with intermittent occurrences of gaps,
duplications, and corrupt data as well as the orbit drifts of satellites (Karlsson et al., 2017). Beginning in
2000, the higher resolution, higher calibration accuracy, and larger number of spectral bands used in the
Moderate Resolution Imaging SpectroRadiometer (MODIS) cloud products resulted in more robust, but
shorter-length products than AVHRR (Kennedy et al., 2012; Claudia, 2012; Stengel et al., 2017),
including MOD08/MYD08 (Marchant et al., 2016) and the Clouds and the Earth's Radiant Energy
System (CERES) (Kato et al., 2018b; Minnis et al., 2011). Meanwhile, the MODIS-based products are
usually reported to overestimate the CF in the Arctic (Trepte et al., 2019; Liu et al., 2022). Although
passive sensor data provide a long time series of continuous CF data covering the entire Arctic region,
the limitations of visible and thermal channels in distinguishing clouds from snow and ice cause the cloud
results of passive sensor data in the high-latitude bright cold polar regions to have questionable accuracy
(Eastman and Warren, 2010; Liu et al., 2010; Liu et al., 2012a; Philipp et al., 2020). Active instruments,
such as CALIOP, do not rely on thermal or visible contrasts in detecting clouds, so they are regarded as
an excellent reference for passive data collection in transient and zonular scenarios (Stubenrauch et al.,
2013; Stengel et al., 2017). However, the number of CALIPSO spatial samplings is too low to overlap
large areas repeatedly in a short time, and the CALIPSO imagers only cover the regions within 82.5°N
latitudes, which greatly reduced spatial and temporal coverages when compared with passive sensor
sensors (Liu et al., 2022; Claudia, 2012; Stubenrauch et al., 2013). Moreover, differences in
instrumentation impose these different cloud definitions, which further increase the biases between the
passive sensor data and the active sensor data. Therefore, an effective method for blending the advantages
of multiple satellite products should yield more accurate Arctic CF products based on a variety of
observations and algorithms.
Several studies have been dedicated to correcting passive sensor data based on active sensor data
with the goal of improving the accuracy of CF products. Philipp et al. (2020) corrected passive sensor
CF data by constructing a function of the sea ice concentration in different seasons and the CF bias in
data acquired from active and passive sensors, which showed reliable results for low-level cloud cover
identification where the sea ice concentration was known (Philipp et al., 2020). Kotarba (2020) matched
the CALIPSO profile data and the MODIS instantaneous field of view to correct passive sensor data
(Kotarba, 2020). This method can be used as an important reference for short-term research that focused
a small area, while the efficiency of the algorithm is also important for the correction of long-time series
and large-scale data. Given that passive sensor CFs exhibit seasonal fluctuations similar to those of active
sensor data (peaking in September and minimizing in April in the Arctic), an approach based on
cumulative distribution function (CDF) matching using time series data may be able to improve both the
accuracy and efficiency of CF detection. Using CDF matching can reduce the systematic bias and root
mean square errors (RMSEs) between target and reference datasets while maintaining the relative
relationship, which has been successfully applied in the study of soil moisture, surface emissivity spectra,
precipitation, and land surface temperature (Drusch, 2005; Brocca et al., 2011; Liu et al., 2011b; Zhang
et al., 2018; Nie et al., 2016; Xu and Cheng, 2021).
In the field of meteorology, to obtain more accurate cloud coverage information, multi-source data
fusion is usually performed based on spectral bands and scale geometry information of instantaneous
satellite images. Examples include various transforms including the contourlet(Miao and Wang, 2006;
Jin et al., 2011), curvelet (Li and Yang, 2008; Liu et al., 2015), NSCT (Wang et al., 2012), and tetrolet
transforms (Zhang et al., 2014). Alternatively, based on the field of view of different observation
instruments used to acquire satellite images and of ground-based stations, methods such as the stepwise
revision method (Kenyon et al., 2016) and data assimilation technology (Hu and Xue, 2007) have been
used. However, in the climate domain, the estimation of a radiative energy budget on a large scale over
a long time series usually requires monthly climate model grid data (Kato et al., 2018a; Sledd and
L'ecuyer, 2021). Using fused instantaneous data to extrapolate climate-scale data may result in a large
accumulation of errors. In recent decades, the fusion of multi-sensor thematic products in climate-scale
studies has been widely used and developed. Two main types of methods exist for merging multiple
satellite thematic products based on the principle of calculation. One type of fusing approach provides
spatiotemporal data fusion by spectral correlation, which is more suitable for the regions where the spatial
information of objects has no obvious change, such as the Spatial and Temporal Adaptive Reflectance
Fusion Model (STARFM) and the improved STARFM (Gao et al., 2006; Hilker et al., 2009; Zhu et al.,
2010; Zhang et al., 2014). The other type of spatiotemporal data fusing method is data-driven, which
involves developing geostatistical models to solve the problem created when the same parameter is
inconsistent among different satellite products. This method includes the Kriging family of techniques
(Chatterjee et al., 2010; Li et al., 2014; Savelyeva et al., 2008), the spatiotemporal interpolation method
(Yang and Hu, 2018), and the Bayesian melding framework (Fuentes and Raftery, 2005; Christakos,
2010). However, these methods rely on Gaussian assumptions and linear models, which limits their
estimation accuracy (Nazelle et al., 2010; He and Kolovos, 2017). A nonlinear spatiotemporal
geostatistical method, Bayesian maximum entropy (BME), has been proposed to fuse the parameters that
have apparent spatiotemporal variations (Nazelle et al., 2010). The BME method can integrate
information from different sources and then consider the data uncertainties in achieving improved

prediction accuracy. The most important advantage of BME is that it does not restrict the complex stochastic relationship between predictions/observations and 'true' values to the Gaussian linearized model; this is an obvious breakthrough over approaches restricted to using normal distributions (Nazelle et al., 2010; Li et al., 2013; Xu et al., 2019). The BME method has broad application in the assessment of many different atmosphere parameters, such as ozone concentration (Nazelle et al., 2010; Bogaert et al., 2009; Christakos et al., 2004), $PM_{2.5}$, $PM_{10}$ (Yu and Wang, 2010; Beckerman et al., 2013), and aerosol optical depth (Xia et al., 2022; Tang et al., 2016). These parameters have similar spatiotemporal properties to CF, i.e., they vary rapidly in both time and space. Therefore, BME has the potential for use in merging multiple satellite CF products to produce spatiotemporally complete, accurate, and coherent Arctic CF products.

In this paper, we present a spatiotemporal data fusion framework based on a CDF matching approach and BME methodology to generate a fused monthly daytime CF product with 1°× 1° resolution in the Arctic region from 2000 to 2020. The CDF matching approach is used to correct the bias of passive sensor data based on active sensor data, thereby improving the quality of the passive data. The BME method is used to produce spatiotemporally complete monthly CF data from corrected multiple-satellite CF products. The uncertainties of passive sensor CF products benchmarked with active sensor data and ground-based data are all considered in the fusing process. The study area was in the Arctic region above 60°N, including land and marine areas. The structure of this paper is as follows. Section 2 describes the data, while Section 3 introduces the data preprocessing and methods. The results and discussion are presented in Sections 4 and 5, respectively. Finally, the conclusions are provided in Section 6.

**2 Data**

**2.1 Satellite Data**

In view of the complementarity among the AVHRR-based, MODIS-based, and active sensor products, this study involved ten passive-satellite-derived products from MODIS and AVHRR, with the time period spanning from 2000 to 2020 along with an active-satellite-derived product from CALIPSO, with the time period spanning from 2006 to 2016. The experimental period only included the sunlit months from April to September because of the darkness of the Arctic winter. All the data are briefly described in Table 1. Our study aimed to provide accurate and reliable measurements of cloud fraction during the daytime in the Arctic region. To achieve this objective, we utilized cloud fraction data labeled as "daytime" from multiple satellite datasets.

The AVHRR sensors are onboard sun-synchronous orbit satellites collecting data in the morning or afternoon (NOAA, Metop-A/B). The morning (afternoon) orbits cross the equator on their descending (ascending) node at approximately 0730 (1330) local time (LT). Starting with NOAA-17 and all MetOp satellites, AVHRR data are available from a midmorning orbit with the equator crossing time at approximately 0930 LT. However, complications arose from changes in the equatorial crossing times of individual AVHRR sensors due to satellite drift (Heidinger et al., 2014; Karlsson et al., 2013). The AVHRR has a nominal spatial resolution of 1.1 km at the nadir point, facilitating full global coverage twice daily (daytime and nighttime), but the products this study employed provide global area coverage data with a nadir footprint size of 1.1 km × 4.4 km (Stengel et al., 2017). Cloud detection algorithms of these latest satellite data have improved greatly in polar regions. However, some data gaps exist as a result of AVHRR scan motor errors (e.g., the NOAA-15 orbits were blacklisted in 2000 and 2001) and

limitations of observation conditions (e.g., CLARA-A2 could not cover the central Arctic Sea in September).

The MODIS sensor is onboard both the morning satellite Terra and the afternoon satellite Aqua, with overpass times at the equator of approximately 1030 LT and 1330 LT, respectively. The MODIS produces complete near-global coverage in less than 2 days. The 36 channels from the visible to thermal infrared spectrum provide abundant spectral information for cloud parameter retrieval. The new version datasets have improved the cloud detection algorithms in polar regions, whereas some researchers found overestimated CF in snow/ice surface in the new datasets when compared with active sensor data (Marchant et al., 2020; Marchant et al., 2016; Paul, 2017; Trepte et al., 2019). Although some differences exist between Terra and Aqua, the consistency between these two satellites cannot be ignored (Trepte et al., 2019).

The CALIPSO satellite combines an active light detection and ranging (lidar) instrument (Cloud-Aerosol Lidar with Orthogonal Polarization - CALIOP Lidar) with passive infrared (Imaging Infrared Radiometer) and visible imagers (Wide Field Camera) to probe the vertical structure and properties of thin clouds and aerosols worldwide (Winker et al., 2007; Vaughan et al., 2004; Hunt et al., 2009; Vaughan et al., 2009; Winker et al., 2009). As the most accurate currently active space-borne instrument for detecting clouds, CALIPSO has a 16-day repeat cycle with equatorial overpass time at 1:30 PM. The CAL_LID_L3_GEWEX_Cloud-Standard-V1-00 is a widely used grid cloud product with a spatial resolution of an equal angle grid 1°×1° (Claudia, 2012).

Table 1. Satellite cloud fraction products used in this research.

| Products | Cloud detect method | Satellite | Sensor | Overpass time | Time range | Temporal resolution | Spatial resolution |
|---|---|---|---|---|---|---|---|
| **MOD08-M3 Terra** | MOD 35 | Terra | MODIS | 1030am | 2000.2-2020.12 | daily | 1°×1° |
| **MYD08-M3 Aqua** | MYD 35 | Aqua | MODIS | 1330pm | 2002.7-2020.12 | daily | 1°×1° |
| **CERES-SSF Terra** | CERES Edition 4 | Terra | MODIS | 1030am | 2000.3-2020.12 | daily | 1°×1° |
| **CERES-SSF Aqua** | CERES Edition 4 | Aqua | MODIS | 1330pm | 2002.7-2020.12 | daily | 1°×1° |
| **CLARA-A2 AM** | EUMETSAT NWC SAF PPS | NOAA-15 | AVHRR3 | 0730am | 2000.1-2000.7 2001.3-2002.7 | daily | 0.25°× 0.25° |
| | | NOAA-17 | AVHRR3 | 0930am | 2002.8-2007.6 | | |
| | | METOPA | AVHRR3 | 0930am | 2007.7-2019.6 | | |
| **CLARA-A2 PM** | EUMETSAT NWC SAF PPS | NOAA-14 | AVHRR2 | 1330pm | 2000.1-2000.12 | daily | 0.25°× 0.25° |
| | | | | | 2001.1-2003.5 | | |
| | | NOAA-16 | AVHRR3 | 1400pm | 2003.6-2005.7 | | |
| | | NOAA-18 | AVHRR3 | 1330pm | 2005.8-2009.5 | | |
| | | NOAA-19 | AVHRR3 | 1330pm | 2009.6-2019.6 | | |
| **PATMOS-x AM** | Naive Bayesian | NOAA-15 | AVHRR3 | 0730am | 2000.1-2000.7 2001.3-2002.8 | daily | 0.1°×0.1° |
| | | NOAA-17 | AVHRR3 | 0930am | 2002.9-2007.6 | | |
| | | METOPA | AVHRR3 | 0930am | 2007.7-2020.12 | | |
| **PATMOS-x PM** | Naive Bayesian | NOAA-14 | AVHRR2 | 1330pm | 2000.1-2001.3 | daily | 0.1°×0.1° |
| | | | | | 2001.4-2003.5 | | |
| | | NOAA-16 | AVHRR3 | 1400pm | 2003.6-2005.7 | | |
| | | NOAA-18 | AVHRR3 | 1330pm | 2005.8-2009.5 | | |
| | | NOAA-19 | AVHRR3 | 1330pm | 2009.6-2020.12 | | |

| | | | | | | | |
|---|---|---|---|---|---|---|---|
| **ISCCP-H AM** | IR and VIS threshold | NOAA-14-NOAA-19; METOPA | AVHRR2 / AVHRR3 | 9000am | | | |
| **ISCCP-H PM** | | | | 1500pm | 2000.1-2017.6 | daily | 1°×1° |
| **CALIPSO-GEWEX** | 5km merged layer product level 2 | CALIPSO | CALIOP | 1330pm | 2006.6-2016.12 | Monthly | 1°×1° |

**2.2 Ground Observation Data**

**2.2.1 Climatic Research Unit Gridded Time Series**

The Climatic Research Unit gridded Time Series (CRU TS) is a widely used climate dataset covering all land surfaces except Antarctica, which uses angular distance weighting to interpolate monthly climate anomalies from extensive networks of weather station observations onto a 0.5° grid (Harris et al., 2020; Harris et al., 2014). This dataset was first published in 2000, and the latest version, CRU TS4.05, contains ten variables including cloud cover for the period 1901–2020 (Harris et al., 2020). The percentage of cloud cover was derived from observations of sunlit hours, and CRU TS4.05 output files are actual values, not anomalies.

**2.2.2 International Comprehensive Ocean-Atmosphere Data Set**

The International Comprehensive Ocean-Atmosphere Data Set (ICOADS) is the most extensive freely available archive of global surface marine data, which has been assimilated into all major atmospheric, oceanic, and coupled reanalysis (Freeman et al., 2017). The ICOADS report is derived from synthetical observations of ships, buoys, coastal platforms, or oceanographic instruments. This dataset offers a gridded monthly summary for 2° latitude × 2° longitude boxes dating back to 1800 (and 1°×1° boxes since 1960) (Woodruff et al., 2005). The available climatic variables include cloud cover and other atmospheric parameters (Bojinski et al., 2014). In this study, we used the 1°×1° cloud cover data in sunlight months (April to September) spanning 2000 to 2020. In particular, we obtained the "fraction of observations in daylight" data from the ICOADS dataset, which allowed us to select only the data points corresponding to daytime observations. During our analysis, we imposed a threshold of 0.8 for the fraction of observations in daylight, ensuring that we only included the data with high confidence in our study.

**2.3 Reanalysis Data and Model Data**

In recent decades, atmospheric reanalysis datasets have emerged as a valuable resource for studying climate processes and predictability, offering a long-term, gridded depiction of atmospheric conditions. These datasets rely on state-of-the-art data assimilation systems, which integrate observational data and underlying models to create a continuous record of historical weather patterns. Through the use of various atmospheric variables, they provide insight into past weather phenomena. The utilization of these datasets could prove imperative in conducting research within areas that are limited in data availability, such as the Arctic. Several studies have investigated the performance of reanalyses over the Arctic for a variety of fields including CF (Yeo et al., 2022; Kennedy et al., 2012; Huang et al., 2017). However, the systematic errors of climatological reanalysis CF are substantial for Arctic clouds because of the complexity of cloud microphysical processes and lack of good observation. In-depth comparisons, as conducted by Walsh, have identified difficulties in adequately depicting persistent low-

level CF in summer via reanalysis models (Walsh et al., 2009).

ERA5 is an advanced atmospheric reanalysis product developed by the European Centre for

Medium-Range Weather Forecasts (ECMWF). It provides information on cloud properties, including
cloud fraction, cloud ice, cloud liquid, rain, and snow water content, which are estimated using the
prognostic equations developed by Tiedtke in 1993(Tiedtke, 1993). This method accounts for physical
processes that act as sources or sinks of clouds, such as convection and condensation. In addition, the
outdated diagnostic temperature-dependent approach for phase partitioning in mixed-phase clouds has
been replaced with a more sophisticated, prognostic method developed by Forbes and Ahlgrimm in
2014 (Forbes and Ahlgrimm, 2014). The updated radiation scheme in ERA5 employs the Monte Carlo
independent column approximation with generalized overlap for sub-grid cloud representation,
enhancing the accuracy of the product.

This study uses the CF of 'ERA5 hourly data on single levels from 1959 to present,' and the CF

parameter has been regridded to a regular lat-long grid of 0.25° and calculated by making assumptions
about the degree of overlap/randomness between clouds at different heights.

The climate model is also a valuable tool for climate studying. However, comparisons of climate

models to Arctic observations over the past three decades have revealed persistent challenges
simulating Arctic climate that partially attribute to imprecise cloud fraction (English et al. 2014). The
sixth phase Coupled Model Intercomparison Project (CMIP6) models have been used in many research
papers about climate. Among them the simulation data of the MRI-AGCM3-2-S climate model
provides a basis for climate research designed to answer fundamental scientific questions and serves as
a resource for authors of the Sixth Assessment Report of the Intergovernmental Panel on Climate
Change (IPCC-AR6). The model employed in this study is derived from the operational weather
prediction model of the Japan Meteorological Agency (JMA). It integrates quasiconservative semi-
Lagrangian dynamics, a radiation scheme, and a land surface scheme that was initially designed for a
climate model. Utilizing observed sea surface temperature (SST) as well as SST alterations forecasted
by atmosphere-ocean coupled models, we carried out simulations of both present-day and future
climate conditions. This model was released in 2017 and provided CF parameters at native nominal
resolutions of 25 km. This resolution employed in the model is as fine as that employed by regional
climate models (RCMs) in recent studies. Smallscale phenomena are realistically simulated in the high-
resolution model, with keeping the same quality of global-scale climate representation as the lower-
resolution models.

The study involved a comparison of pre- and post-fusion CF data with reanalysis and model data.

The aim was to underscore the significant role of fused data in improving the consistency of CF
between satellite observations, reanalysis data, and model data.
**3 Data Preprocessing and Methodology**

In this study, we propose a fusion algorithm framework that combines data from multiple satellites

to provide CF datasets with high spatiotemporal coverage and improved accuracy. Figure 1 shows a
flowchart of the general process, which includes four parts. First, the original data were preprocessed
before data fusion, a process that included data quality control and data resampling. Second, bias
correction of passive sensor data was conducted using active data with the CDF matching method. Third,
to comply with BME's stationarity prerequisite that assumes constancy of mean and variance, we
removed the spatiotemporal trend of the original satellite CF data over the study area using the

spatiotemporal moving window filter method. Fourth, the spatiotemporal covariance function was modeled based on the isotropic residual data, and then the entropy was maximized with covariance constraint. All the satellite-based CF data were treated as soft data so that the associated uncertainties were incorporated into the fusing process.

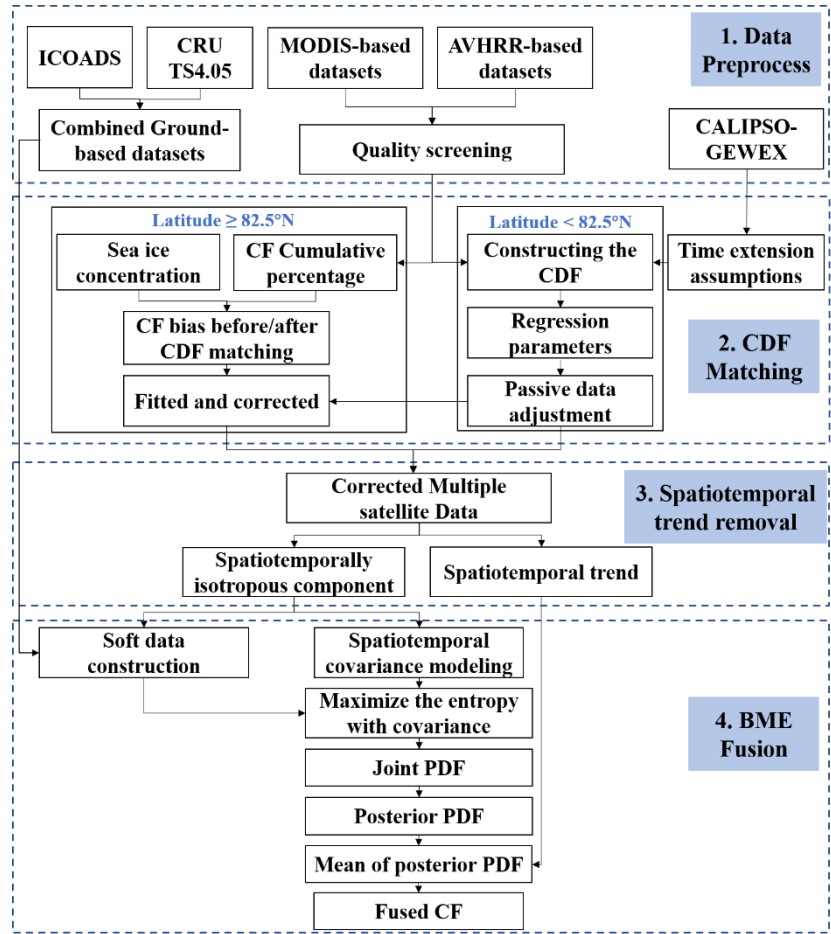

Figure 1. Flowchart for merging the multiple satellite cloud fraction products based on cumulative distribution function matching and the Bayesian maximum entropy method.

## 3.1 Data Preprocessing

Over the Arctic, the cloud detection capabilities of passive sensors are always limited by spectral channels, while active sensors are not susceptible to these effects (Liu et al., 2010; Liu et al., 2012b; Kotarba, 2020; Shupe et al., 2013). To obtain more accurate fused CF results, it is necessary to correct these passive sensor products using active sensor data before merging.

For satellite datasets, statistics always have the Scientific Data Set (SDS) name suffix "_Standard_Deviation" and which are computed by calculating an unweighted standard deviation of all pixels or samples within a given 1° grid cell. The large CF standard deviations (STDs) of satellite datasets represent the large uncertainties of CF detection (Ackerman et al., 2008; Stengel et al., 2017). In this study, we calculated the relationship between differences in STDs and CFs of passive/active sensor datasets and found that the larger the standard deviation, the more serious the underestimation of passive sensors. For the products with standard deviation flags, including MOD08 Terra/Aqua, CLARA-A2 AM/PM, and the PATMOS-x AM/PM, we used the 90 percentile of the daily standard deviation as scene-

based dynamic thresholds to screen CF data.
However, no standard deviation information was available for CERES-SSF Terra/Aqua and the
ISCCP-H AM/PM datasets. Based on research that shows ignoring optically very thin clouds could
increase the agreement between passive sensor data and the CALIPSO data, the 0.15 COT dataset was
selected as the quality threshold in this study.

**3.2 CDF Matching**

A widely used scaling strategy known as CDF matching can be used to adjust the distribution of the
target dataset to the range of reference data under the constant relative relationship. Several studies have
proved that the process of adjusting this distribution does not change the variation of original satellite-
based products, but rather aligns the value range with that of the reference data (Liu et al., 2011b; Brocca
et al., 2011; Xu and Cheng, 2021). Based on similar seasonal fluctuations of the passive sensor CFs and
active sensor data, the time series of passive sensor data from each grid box in the Arctic region were
adjusted to the values of the paired CALIPSO-GEWEX latitude and longitude grid. However, the
CALIPSO-GEWEX data could not cover regions with a latitude greater than 82.5°N and the temporal
range only covers 2006–2016. To correct the CF bias over the entire Arctic region, two strategies were
considered.
First, for the regions with enough reference data, the CF data of all passive sensors were directly
adjusted by CDF matching. The matching approach includes three steps: (1) constructing the cumulative
distribution function, (2) deriving regression parameters, and (3) adjusting the original data with
regression parameters. In our study, we use a three-month moving mean to eliminate the uncertainties in
CALIPSO-GEWEX data caused by the limitation of sampling quantities and frequencies. The filtered
daily passive sensor datasets were resampled as monthly mean data, and then the CDFs were constructed
for every dataset based on the same method used for the active data. A least-square fit was used to derive
the relationship between the reference and the target datasets. Based on the analysis of Liu et al.(2022),
the seasonal variation of CF for multiple satellites was greater than the interannual changes in CF (Liu
et al., 2022). We propose an additional assumption that the CDF ratio between active and passive sensor
data remains constant over the years in a 1°×1° grid cell.
Second, it was difficult to implement the CDF matching strategy for areas beyond the coverage of
active sensor data. Considering the relationship among the CF bias before and after CDF correction, the
cumulative percentage of CF (CPCF, the average CF over an interval of SIC), and the sea ice
concentration (SIC), a fitting function is proposed to correct the CF data.
After executing the abovementioned steps, we obtained the corrected multiple satellite data.

**3.3 Spatiotemporal Trend Analysis and Removal**

The BME theory was constructed based on the hypothesis of spatiotemporal random field (S/TRF)
(Nazelle et al., 2010; Christakos, 2000; He and Kolovos, 2017), which means that all the variables used
for this process are homogeneous and isotropous. However, a natural process that evolves in space–time,
such as the distribution of CF, can be divided into a heterogenetic global spatiotemporal trend and a
spatiotemporally isotropous residual, following Eq. (1):

$$CF_{(s,t)} = \overline{CF}_{(s,t)} + CF_{res(s,t)},\tag{1}$$

where (s, t) represents the space and time, $\overline{CF}_{(s,t)}$ represents the global spatiotemporal trend, and $CF_{res(s,t)}$
represents the stochastic anomalies of the variable. To meet the second-order stationarity assumption
(constant mean and variance), it is necessary to remove the global spatiotemporal trend before estimating
the spatiotemporally autocorrelated structure of the data (Spadavecchia and Williams, 2009; Tang et al.,
2016). In this study, the global spatiotemporal trend was calculated using a spatiotemporal filter window
with a size of 5° (longitude) × 5° (latitude) × 3 (months).
Figure 2 shows a histogram of the original combined satellite CF data, the global spatiotemporal
trend, and the residual spatiotemporally isotropous component. From these distributions of the histogram,
the residual is approximately normally distributed, which meets the requirement for modeling the
structure of the spatiotemporal autocovariance.

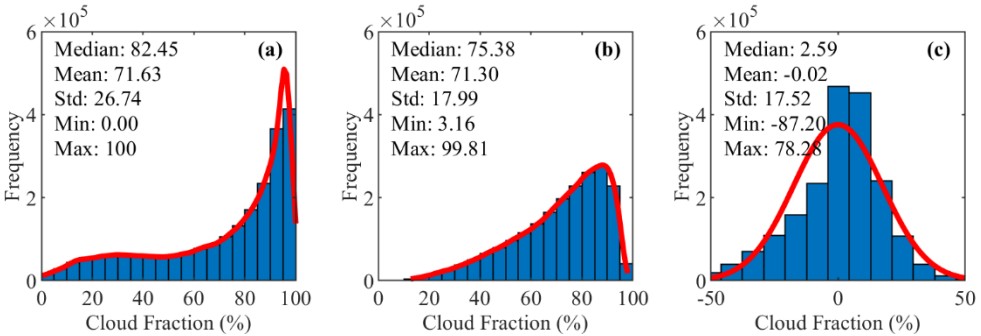

Figure 2. Histograms of (a) original combined satellite cloud fraction, (b) global spatiotemporal trend, and (c)
spatiotemporally isotropous component, for the entire Arctic area (Example using 2010 data).

**3.4 BME Fusion**
**3.4.1 Spatiotemporal Covariance Modelling**
In spatiotemporal geostatistics, a covariance function indicates the spatial and temporal dependency
of the data, which decreases as distance/time increases (Griffith, 1993). The spatiotemporal variation of
the CF also can be expressed by a spatiotemporal covariance function. In the BME method, the
experimental covariance can be calculated from the point pairs at specific distances and then modeled by
the commonly used covariance model (Cressie, 2015). This study uses a nested covariance model with
two spatiotemporal exponential models to model the spatiotemporal covariance of the detrended
combined CF data, following Eq. (2):
$$\text{cov}(d,\tau) = c_1 \exp\left(-\frac{3d}{a_{s1}}\right)\exp\left(-\frac{3\tau}{a_{t1}}\right) + c_2 \exp\left(-\frac{3d}{a_{s2}}\right)\exp\left(-\frac{3\tau}{a_{t2}}\right), \qquad (2)$$
where $d$ is the spatial lag and $\tau$ is the temporal lag between point pairs at coordinates $p(s, t)$ and
coordinates $p'(s', t')$; $c_1$ and $c_2$ are the partial sill variances of the two exponential models; $a_{s1}$ and $a_{s2}$ are
the spatial ranges of the two exponential models; $a_{t1}$ and $a_{t2}$ are the temporal ranges of the two exponential
models. When the S/TRF is characterized by spatial and temporal stationarity, it is only the relative
distance between any couple of locations that affects the covariance function. Specifically then, the
covariance function has the same value $cx(p, p') = cx(r, t)$ for any location pair $(p, p')$ separated by the
same spatial distance vector $r = s' - s$ and same temporal distance lag $\tau = t' - t$ (Christakos and Serre 2000).
In this study, the parameters for spatiotemporal covariance are modeled separately for each year. The
modelled results shown that the model has a spatial range of 2°, a temporal range of 3 months, and a
partial sill variance of 0.85 for local scale CF (the first nested covariance model). And for the large range
CF the model has a spatial range of 30°, a temporal range of 6 months, and a partial sill variance of 0.15
(the second nested covariance model).

**3.4.2 Construction of Soft Data**
BME treated the informative content with uncertainty from different sources as soft data (He and
Kolovos, 2017). For example, the observed data that accompanied by obvious sources of uncertainty
such as inaccuracy in measuring devices, modeling uncertainties, and human error. In this study, the CF
data of passive sensor products are viewed as soft data. For the BME method, a key conceptual aspect is
that the framework does not impose any restrictive assumptions about the PDFs of soft data. Hence, a
parameterized statistical distribution of different sources of information can be used to replace the real
PDFs (Nazelle et al., 2010). Soft data could be probabilistic or interval soft data (Christakos, 2000). In
this study, the differences between satellite data and ground observations followed normal distributions
approximately. Therefore, the passive sensor data used for fusion were all treated as soft data with a
Gaussian distribution, following Eq. (3):
$$CF_{sate,x} = CF_{ground,x} + \varepsilon_x, \qquad (3)$$
where $CF_{sate,x}$ and $CF_{ground,x}$ are the satellite CF data and the corresponding ground observation,
respectively, and $\varepsilon_x$ is an independent random error, following Eq. (4):
$$\varepsilon \sim N\left(\mu_\varepsilon, \sigma_\varepsilon^2\right), \qquad (4)$$
where $\mu_\varepsilon$ represents the mean of random error and $\sigma_\varepsilon^2$ represents the variance (Tang et al., 2016).

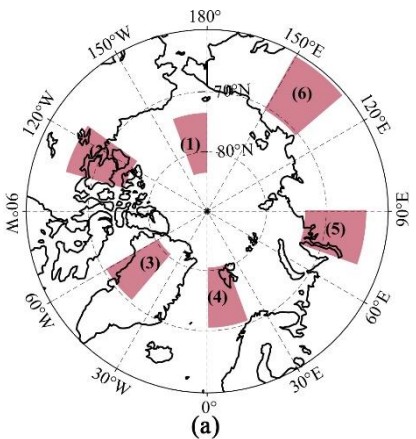

(a)

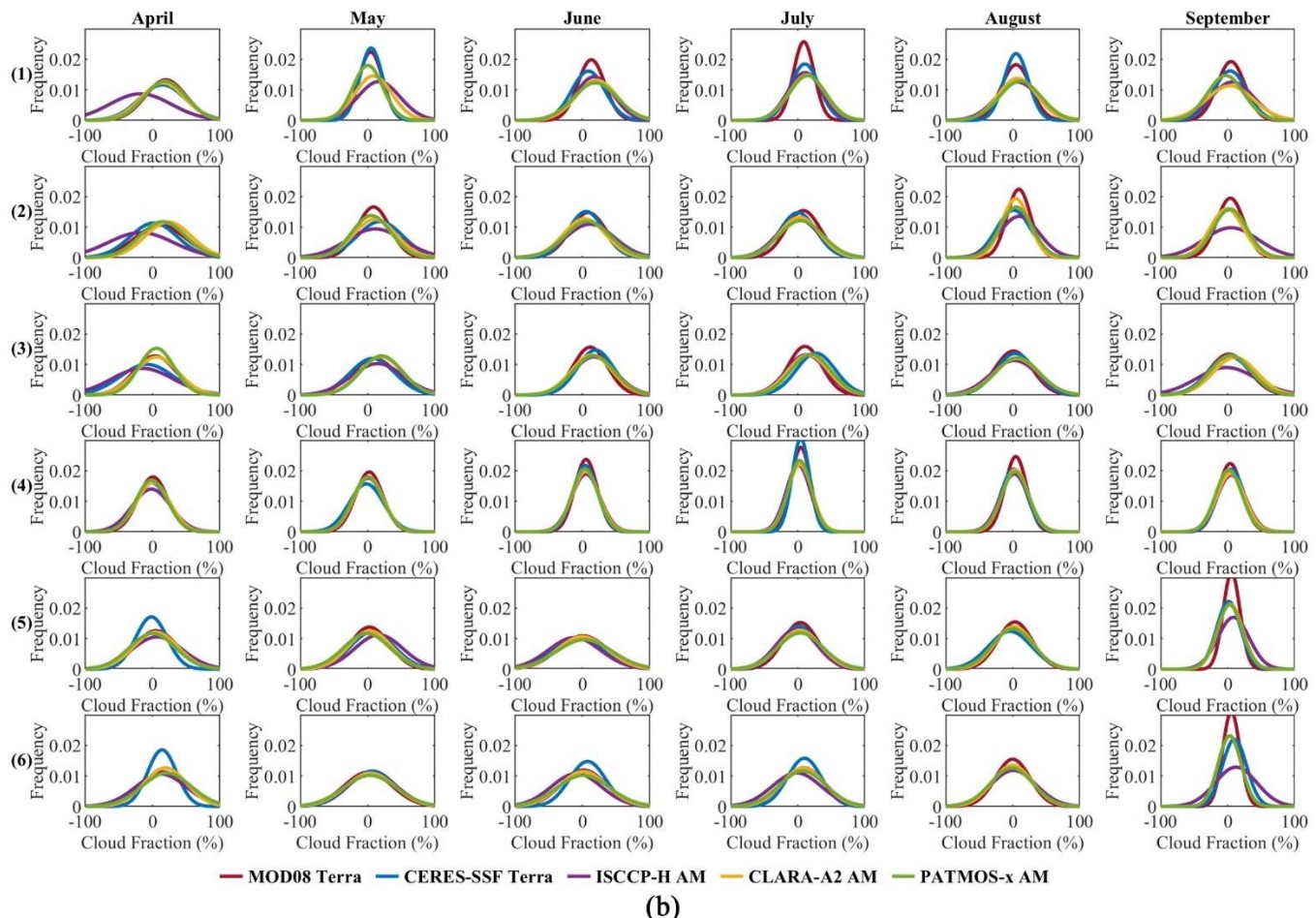

Figure 3. Gaussian probability density functions of the random errors between each type of satellite data and ground observations at six randomly selected regions of interest from April to September.

Because the uncertainties in each satellite CF data vary at different spatial and temporal scales, using the average uncertainty of the entire dataset to construct soft data over the entire study area will undoubtedly neglect the spatiotemporal variation of uncertainties. In this study, six regions were randomly selected to analyze the probability density functions (PDF) of random errors (Fig. 3). Large inconsistencies were observed for the PDF in land and ocean regions, and the temporal variation was also an important factor in inconsistencies. We constructed the soft data for CF data over land and ocean regions in every month separately. Considering the large errors in the Greenland Ice Sheet (GrIS), we calculated the PDF of random error separately for that region.

For each grid box, the CFs of different satellite data were converted into a Gaussian distribution probability soft data, individually (Tang et al., 2016). The soft data were expressed as:

$$CF_{soft,sate} \sim N\left(CF_{sate} + \mu_\varepsilon, \sigma_\varepsilon^2\right),\tag{5}$$

where $CF_{sate}$ is the detrended CF value of multiple satellite datasets; the mean and variance of the Gaussian distribution probability soft data were expressed by $CF_{sate}+\mu_\varepsilon$ and $\sigma_\varepsilon^2$, respectively.

**3.4.3 Using the BME Method for Multiple CF Data Fusion**

The BME method can be used to merge continuous variables of satellite data for some atmospheric

parameters. To simplify the heterogeneity and anisotropic variability, the residuals were considered only in the fusion process. Assuming that various adjacent observations from satellites were available with irregular spatial and temporal gaps, the nonlinear mean estimation $\overline{x_k}$ of CF at the location $(s_x, s_y)$ at time $t$ was estimated as:

$$\overline{x_k} = \int x_k f\left(x_k \mid x_{soft,1}, x_{soft,2}...x_{soft,n}\right) dx_k, \tag{6}$$

where $f(x_k \mid x_{soft,1}, x_{soft,2}...x_{soft,n})$ is a posterior PDF over the spatiotemporal adjacent grid observations, and $x_{soft,1}, x_{soft,2}...x_{soft,n}$ are the probabilistic Gaussian soft data derived from multiple satellite data. The posterior PDF at the estimation point updates from the prior PDF in the Bayesian rule when soft data are involved, so the relationship can be expressed as:

$$f\left(x_k \mid x_{soft,1}, x_{soft,2}...x_{soft,n}\right) = \frac{f\left(x_{soft,1}, x_{soft,2}...x_{soft,n}, x_k\right)}{f\left(x_{soft,1}, x_{soft,2}...x_{soft,n}\right)}, \tag{7}$$

where $f(x_{soft,1}, x_{soft,2}...x_{soft,n})$ represents the prior PDF of the spatiotemporally isotropous CF at the adjacent grid, $f(x_{soft,1}, x_{soft,2}...x_{soft,n}, x_k)$ is the joint PDF without specific information. Generally, the joint PDF is represented by $f_g(x_{map})$, which can be calculated by maximizing the entropy under the constraint of the general knowledge $g$ (Jaynes, 1957). When predicting the probability distribution of a random event, the larger the information entropy, the larger the amount of information obtained, and the result is closer to the actual situation under a most uniform probability distribution. In this study, general knowledge is the spatiotemporal covariance model, and to maximize the entropy, we introduce a Lagrange multiplier $\lambda$ (Xia et al., 2022).

$$f_g\left(x_{map}\right) = \frac{\exp\left(\sum_{\alpha=1}^{n} \lambda_\alpha g_\alpha\left(x_{map}\right)\right)}{\int \exp\left(\sum_{\alpha=1}^{n} \lambda_\alpha g_\alpha\left(x_{map}\right)\right) dx_{map}}, \tag{8}$$

Finally, the expectation of spatiotemporally CF isotropous component can be calculated by solving these equations. Then the anisotropic spatiotemporal trend component of each grid was added to the expectation at the corresponding point to obtain the merged CF product.

## 4 Results

### 4.1 Result of CDF Matching

Figure 4 shows the scatter plots of the CF distribution before and after CDF matching from multiple passive and active sensors at the valid grid boxes with a latitude of less than 82.5°N. Based on the fact that the assumption that the correction coefficient does not vary over time, the training datasets (T) were processed from 2008 to 2014 and the validation datasets (V) were processed in 2006, 2007, 2015, and 2016. In Fig. 4, the 'Original CF (T)' and 'Original CF (V)' indicate the comparison of CALIPSO-GEWEX CF and that of the original passive sensor data, so that the 'CDF CF (T)' and 'CDF CF (V)' represent the comparison between CALIPSO-GEWEX CF and the corrected CF. In general, for all the passive sensor datasets, the CFs after CDF matching were closer to the 1:1 line than before CDF matching. $R^2$ increased by about 0.07–0.15, while that for ISCCP-H products was over 0.45. The RMSEs decreased to one-third to one-half of what they were, and the biases decreased to approximately zero, which means that the CDF matching obviously corrected outliers and eliminated the average differences between the

passive and active sensor CFs. From these scatter plots, we also understand that CDF matching plays an important role in low CFs (less than 60%), which was always seen in April or on the GrIS(Liu et al., 2022).

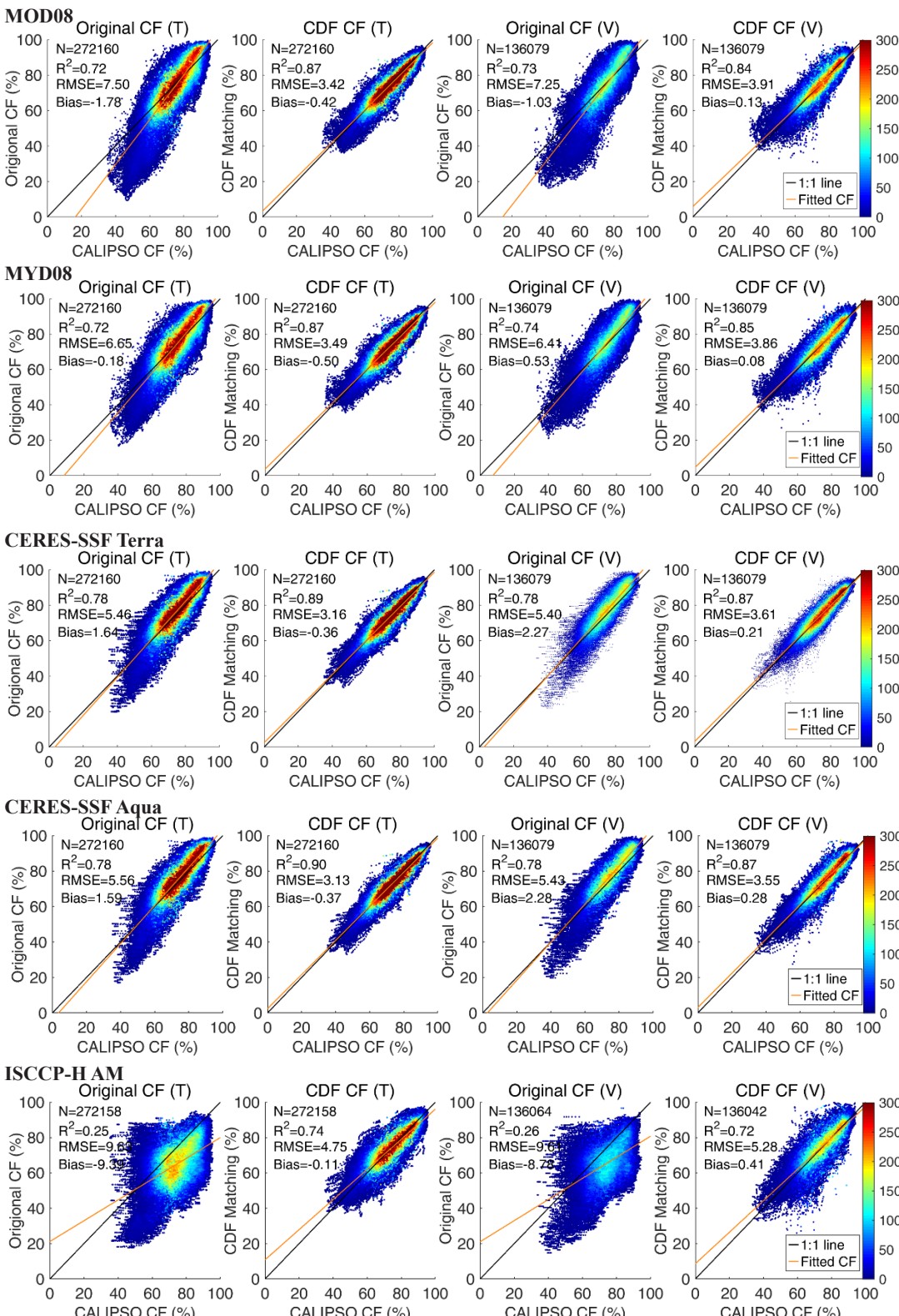

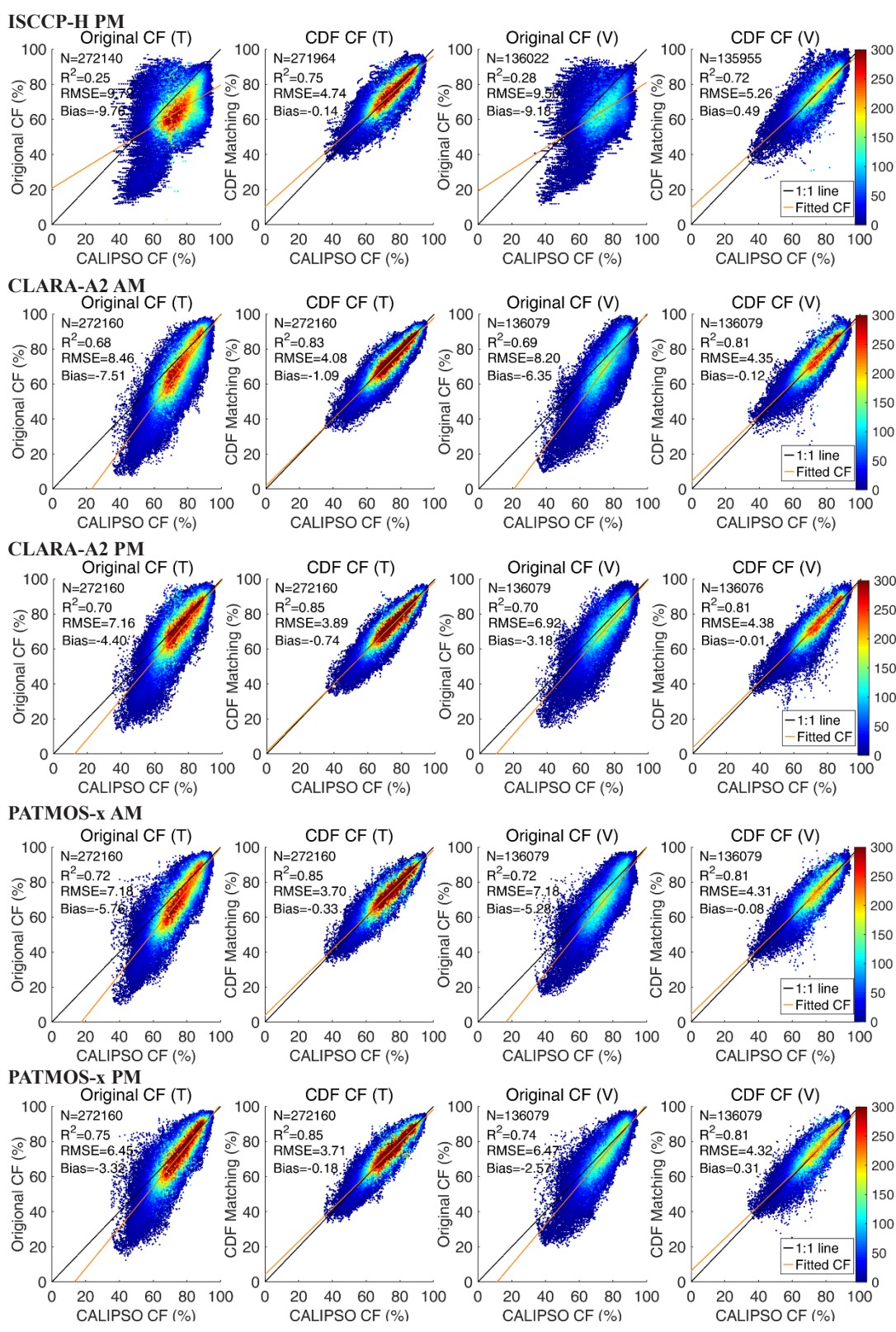

Figure 4. The scatter plots of the cloud fraction comparison between the passive and active sensor datasets at regions with latitudes less than 82.5°N before and after cumulative distribution function matching: (T) means training data with time ranges from 2008 to 2014 and (V) means validation data from 2006, 2007, 2015, and 2016.

In the sea ice regions, the relationships between CF bias of passive sensor data after and before CDF

matching, CPCF, and SIC are shown in Fig. 5. The results indicated that the mean of bias increased with

the SIC. Moreover, the CPCF appeared to decrease with increasing SIC, a negative correlation between

CPCF and bias was also evident.

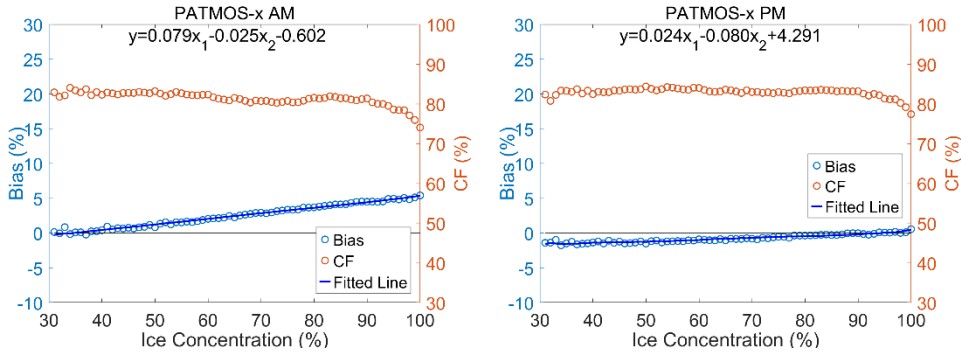

Figure 5. The relationship between cloud fraction bias of passive sensor data after and before cumulative distribution function matching, the cumulative percentage of cloud fraction, and the sea ice concentration in sea ice regions with latitude less than 82.5°N.

By virtue of this association, SIC and CPCF are modeled as dependent variables of the bias. Due to the predominant presence of sea ice over the domain located above 82.5N, we employ this functional association to remediate CF inaccuracies in the region, called C-SIC Corrected CF. Figure 6's initial two panels depict a comparison between the CF of active data and passive data before and after correction by C-SIC in sea ice regions below 82.5°N. The results indicate that $R^2$ of the corrected scatter plots increased slightly, but the RMSEs and bias were greatly reduced. In particular, the CF underestimated by passive sensors was similar to that of active sensors after correction. In our previous study, we have proven that this type of underestimation is very common(Liu et al., 2022). The third panel of Figure 6 shows the comparison of C-SIC Corrected CF and the CDF matching CF in sea ice regions with latitude less than 82.5°N. The results also showed that the C-SIC Corrected CFs have high degree of consistency with the CFs corrected by the CDF matching, with $R^2$ over 0.75, RMSE less than 3.6, and bias less than 0.5. However, although the correction has improved the ISCCP-H CFs, they also showed large inconsistencies with the passive sensor data and the CDF matching data. Therefore, the ISCCP-H CFs in regions north of 82.5°N were not included in the following fusion process.

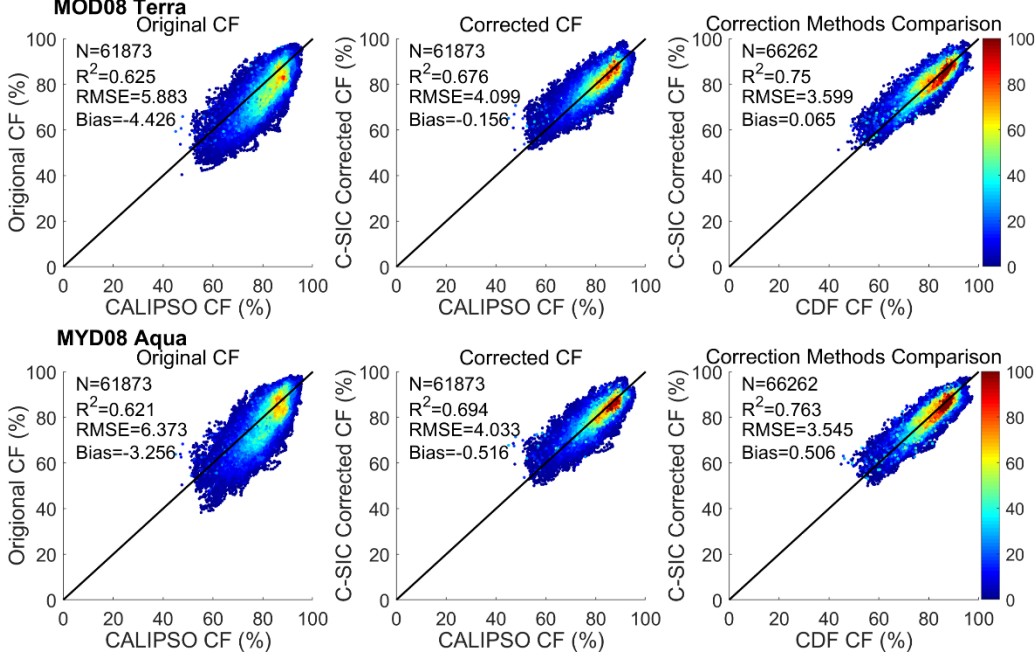

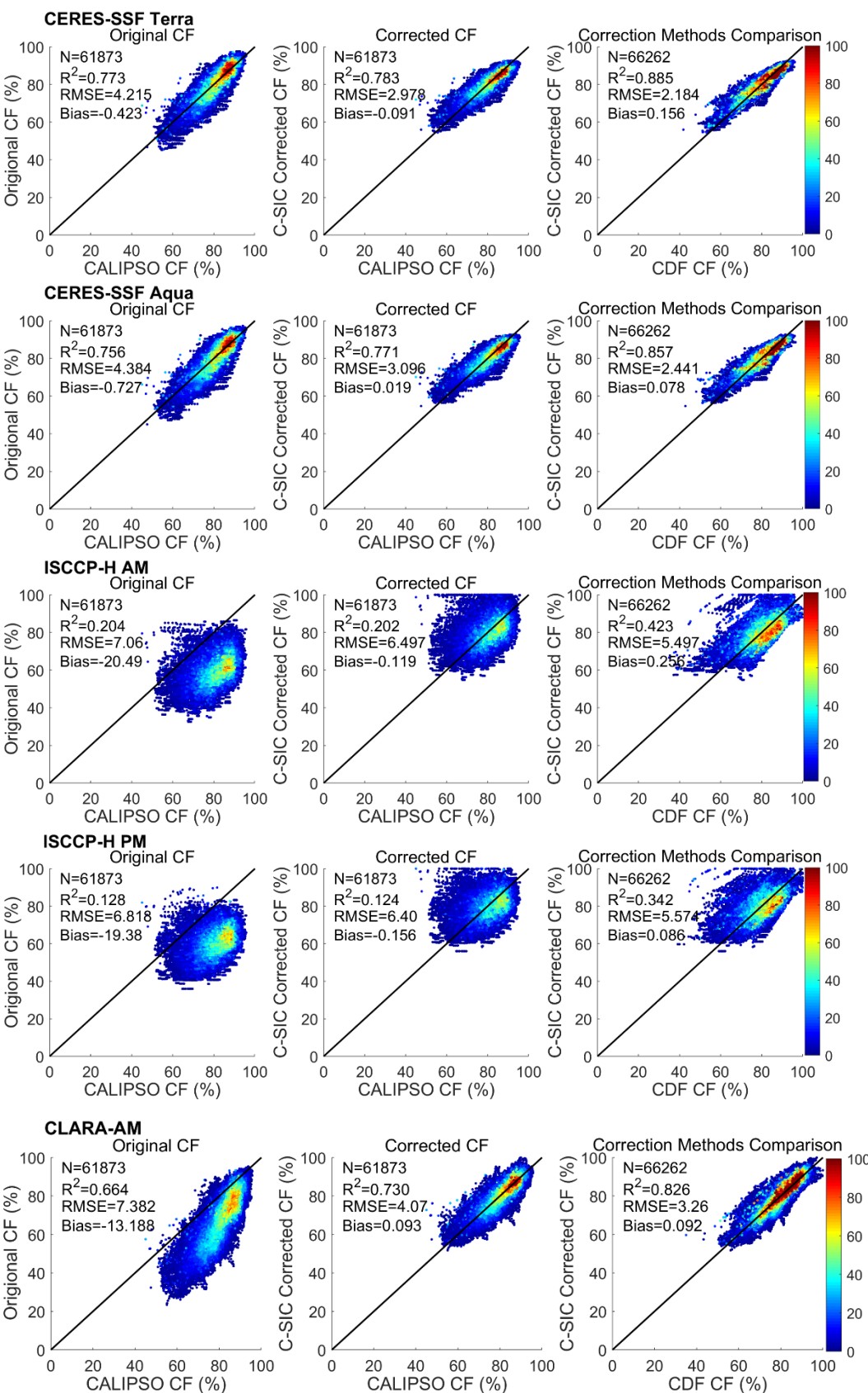

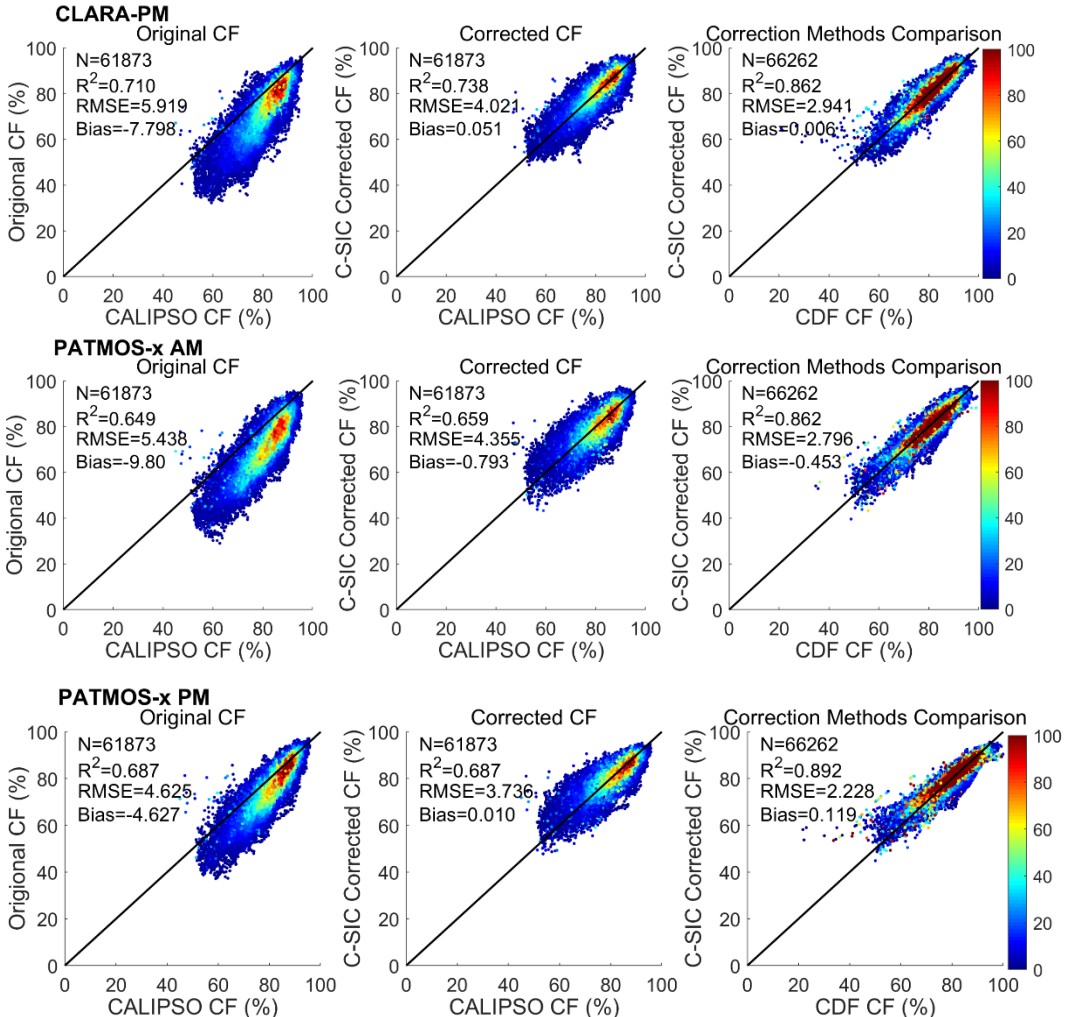

Figure 6. The scatter plots of the cloud fraction (CF) comparison between the passive sensor datasets and the
active sensor dataset before (the first panel) and after (the second panel) using the method of CF corrected by the
cumulative percentage of CF and SIC (C-SIC). And the scatter plots of the results comparison between C-SIC and
cumulative distribution function matching (the third panel).

Accompanying the decreases in the CF differences of the active and passive sensor data, the
accuracy of individual passive sensor datasets for the entire Arctic during the experimental period was
also generally improved. Moreover, the consistency of multiple satellite data has improved greatly.
Figure 7 displays the standard deviation between $1° \times 1°$ passive sensor CF data before and after the
application of cumulative distribution function matching (latitude≤82.5°N) and C-SIC correction
(latitude >82.5°N). The results obtained from different regions indicate an obvious decrease in the
inconsistency between multiple passive sensor data after the correction with the aforementioned methods.
In the Holarctic region, multiple passive sensor CFs saw a decrease in mean STD from 9.18% to 5.75%,
with more than 50% of the corrected data displaying a standard deviation within 5%. The sea ice region
saw the largest reduction rate of the mean STD, approximately 4.5%. This reduction was mainly derived
from a STD value range of 10–15%, due to the limited detection capacity of passive sensor data in sea
ice areas. Regions with latitude less than 82.5°N saw a decrease in mean STD of only 3.02%. In contrast
to the sea ice region, these land regions saw a smaller standard deviation between multiple satellite data.
The distribution of STD frequency in regions over 82.5°N and the entire sea ice area appeared similar,
indicating that the C-SIC correction method was highly effective in 82.5°N regions. Although the relative
values showed improvement, the absolute change appeared inconspicuous.

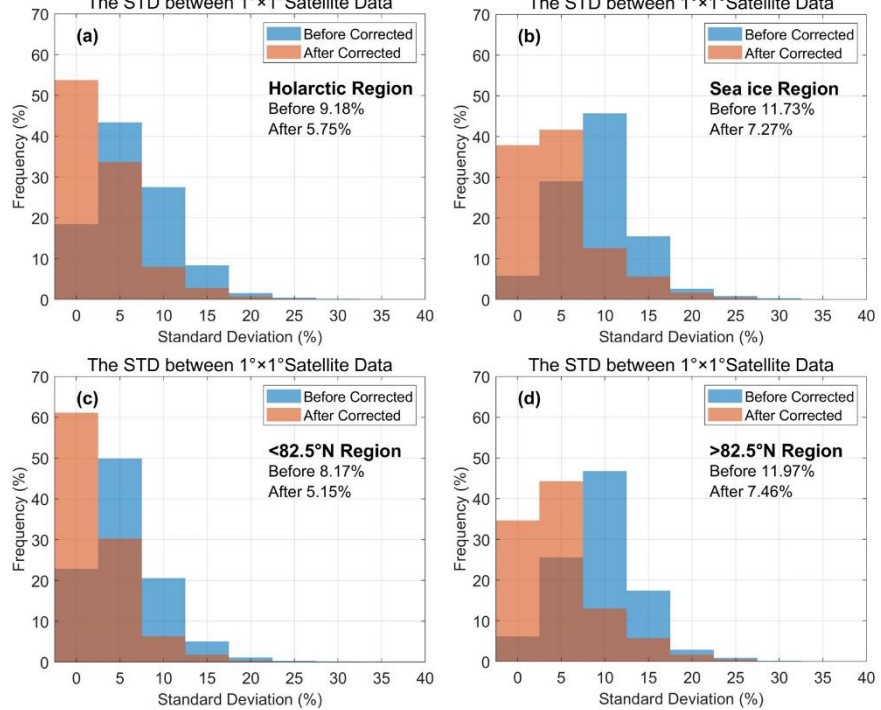


Figure 7. Standard deviation between 1° × 1° passive sensor cloud fraction before and after cumulative distribution

function matching (latitude<82.5°N) and C-SIC Corrected (latitude >82.5°N).


**4.2 Result of BME Fusing**
**4.2.1 Spatial and Temporal Distribution of the Fused CF**

Figure 8 shows the spatial distribution of Arctic CF from the fused product, multiple satellite data,

and ground observations. The results indicate that although most satellite-based products agreed
relatively well with the ground-based observations in both the geographical distribution and the zonal
average of Arctic CF at first glance, large disparities also appeared in some specific regions, whereas the
fused product we proposed reduced these disparities apparently. For instance, nearly all the passive and
active sensor products show the CFs over the GrIS were less than 60%. However, CFs of ground-based
observations over this region were reported as nearly 70%, which is closer to that of the fused product.
The sea regions of the central Arctic, which are covered by perennial sea ice/snow, are another area where
the passive sensor products always underestimate CF. From these figures, some passive sensor products,
especially for the AVHRR-based datasets, have CFs that are about 10–20% lower than those of active
sensor data and ground-based observations. However, the fused CF has a similar magnitude to these two
referred datasets.

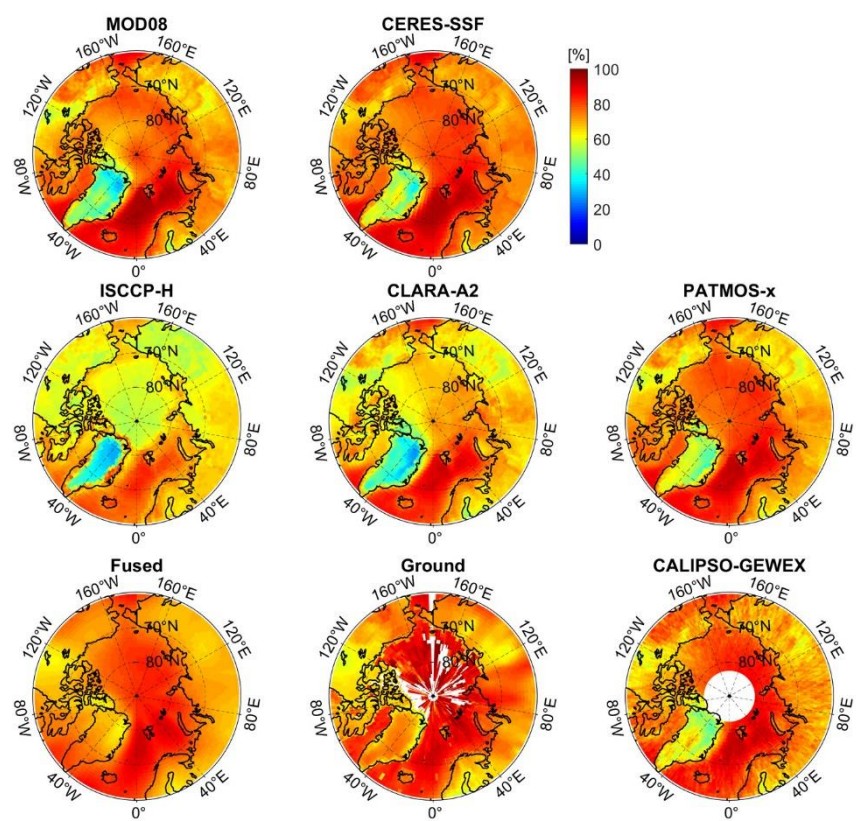


Figure 8. Distribution of the average cloud fraction of different datasets over the Arctic from 2000 to 2020. The
time ranges for ISCCP-H and CALIPSO-GEWEX were from 2000 to 2017 and from 2006 to 2016, respectively.

By contrast, the ground-based CF products have a large data gap because ground weather stations
are sparsely distributed in the Arctic, so the limitation of sampling quantities and frequencies had the
effect of limiting the spatial and temporal ranges of active sensor data. Moreover, the AVHRR-based
products often suffer from missing data as a result of satellite failures or band switching (Hollmann,
2018); in addition, some passive sensor products such as CLARA-A2 have some spatial gaps over the
Arctic Sea during autumn (Karlsson et al., 2017). Although we have eliminated a large number of low-
precision daily data in preprocessing, the completeness of the merged multiple-satellite CF products is
obviously higher than those of the original satellite-based data and ground-based observations in both
spatiality and temporality, especially in regions of the Arctic Ocean. The spatial completeness (the ratio
of available data to the CF grids of the entire Arctic) of the fused CF product was nearly 100%, which is
much larger than 54.09% of ground-based products and 73.15% of the active sensor product. Therefore,
the fusion algorithm proposed by this study can not only obviously reduce the inconsistencies of Arctic
CF between multiple satellite products and reference datasets but also effectively compensate for the data
gaps caused by the lack of reference data.
It is well known that the CF in the Arctic regions fluctuates apparently with the change in seasons.
To show the temporal accuracy of the fusion products, we analyzed the long time series area-weighted
mean of the CF. Figure 9 depicts the fluctuation of the mean value on a monthly basis for all data
during sunshine periods (April to September) before and after fusion, as demonstrated by the time
series. It is clear that the CF peaks in September and reaches a minimum in April. However, only the
fused product always maintains a high level of consistency with the reference data, with the monthly

mean CF varying from 62% to 79%. The overall area-weighted mean of the differences between fused CF and CALIPSO-GEWEX CF and between fused CF and ground-based CF was about 0.91% and 0.40%, respectively, which are about one-third of the differences for MODIS-based products and reference products and about one-fifth to one-twentieth of the differences for AVHRR-based products and reference products. In land and ocean areas, the fusion algorithm clearly corrects the outliers with large deviations, such as the CF from CLARA-A2, PATMOS-x products, and the CERES-SSF products. The first two datasets are well-known for underestimating the Arctic CF dramatically (Karlsson et al., 2017; Karlsson and Dybbroe, 2010). In this study, the underestimation mainly occurred in April, with approximately 8% and 3% for those two datasets, respectively. The latter has often been reported to overestimate CF (Doelling et al., 2016; Trepte et al., 2019), and in this study the CERES-SSF products nearly overmeasure CF all year long from April to September. However, the fusion framework proposed by this study scales these underestimated values or overestimated values to a range similar to that of active sensor data by CDF matching; meanwhile, it takes into account the deviation from ground observations in the BME fusion process. The fused CFs can not only reduce the overestimation of CF by MODIS-based products but also decrease the underestimation of CF for AVHRR-based products, which obviously improves the consistency of CF between the active sensor, passive sensor, and ground observation dataset compared with the original data.

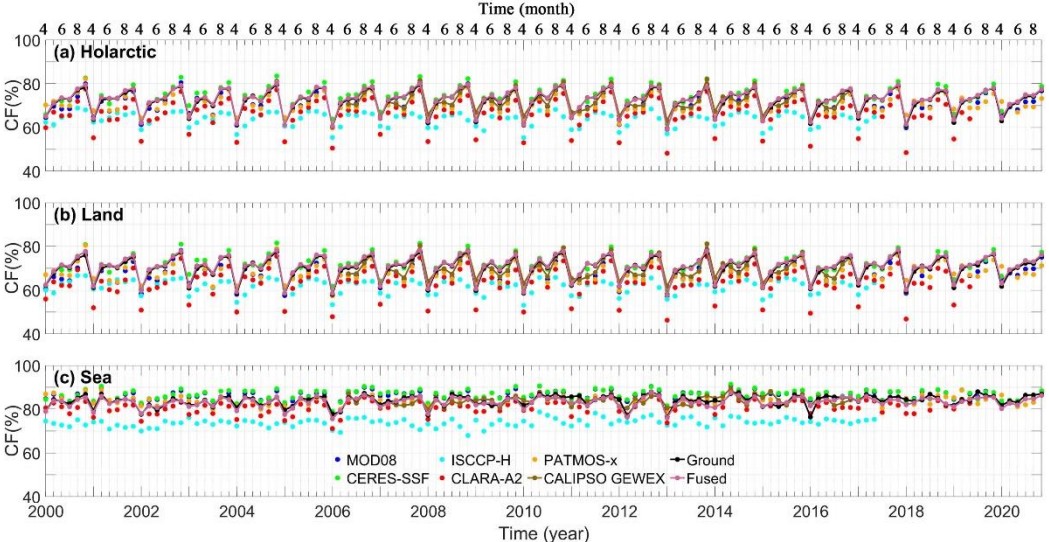

Figure 9. The area-weighted means of cloud fraction over (a) Holarctic, (b) Land, and (c) Sea for different products in the Arctic from April to September during 2000 to 2020. The time ranges for ISCCP-H and CALIPSO-GEWEX were from 2000 to 2017 and from 2006 to 2016, respectively.

**4.2.2 Quantitative Assessment of Fused CF**

To validate the fused CF and compare the accuracy of the fused results to that of several original satellite CFs, all the passive sensor CF products and the merged CF product were spatiotemporally compared with the CRU TS4.05 in land regions and ICOADS measurements in sea regions. The correlation coefficient ($R^2$), root-mean-square error (RMSE), and mean bias (bias) were used to quantitatively evaluate the accuracies of the original and merged CF products. As Fig. 10 indicates, the scatters of the fused CF product and ground-based observations were closer to the 1:1 line than that of the original satellite data. In this case, the fused data had the largest $R^2$ (0.51), lowest RMSE (6.95%),

and the lowest bias (0.35%) for land regions. In addition, the fused data had the largest $R^2$ (0.42), the
lowest RMSE (5.62%), and the lowest bias (0.55%) for sea regions.

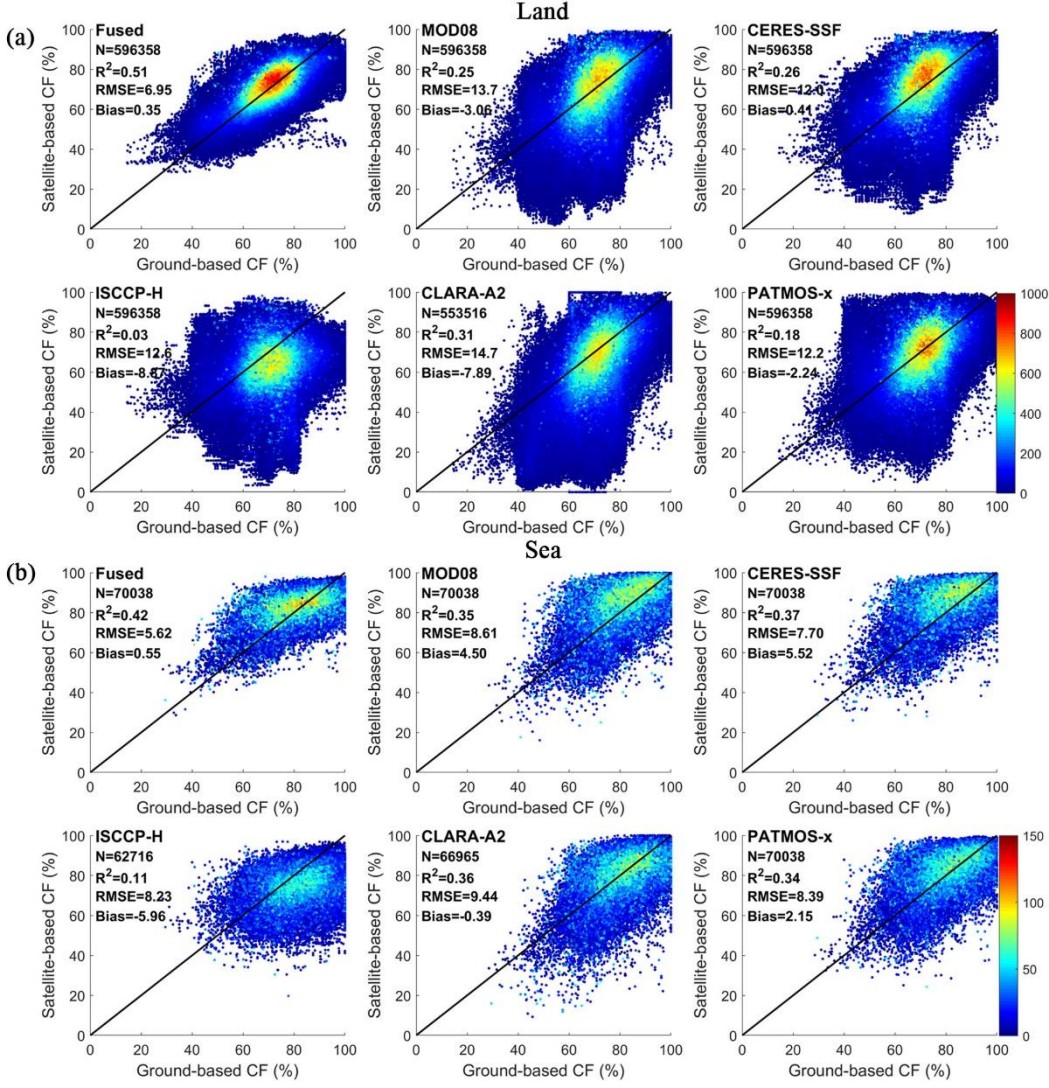

Figure 10. Validation of the fused cloud fraction and the original passive sensor datasets against the (a) CRU
TS4.05 and (b) ICOADS datasets.

For land, it can be also seen that the fusion results have a strong ability to correct the satellite CF
that is less than 30%. These values were mainly found on the GrIS, in the Canadian Islands, and on the
central Eurasian continent. In addition, the RMSE of CF after fusion was only one-half of the original
satellite data, which means that the overall distribution of the fused CF is better fitted to the reference
data, and most of the CFs with differences over 30% were well-corrected.
The observations of ICOADS come from multiple observation platforms, and most of these
platforms operate in open waters. The open water regions varied mostly with the growth and decline of
the SIC, which brings great spatiotemporal heterogeneity for the sampling of ICOADS. Therefore, in the
verification process, the first step was to spatiotemporally collocate the satellite data with ocean site.
Figure10 (b) shows that $R^2$ of the fused CF only improved by about 0.05–0.08 when compared with most
satellite data. However, the fusion algorithm reduces the RMSEs and bias obviously. The RMSEs of the
fusion CF were about one-fourth to one-third of the original MODIS-based products and one-third to
three-fifths of the original AVHRR-based products. The reductions of bias were about 4–5% for MODIS-
based products and about 2–5.4% for AVHRR-based products.

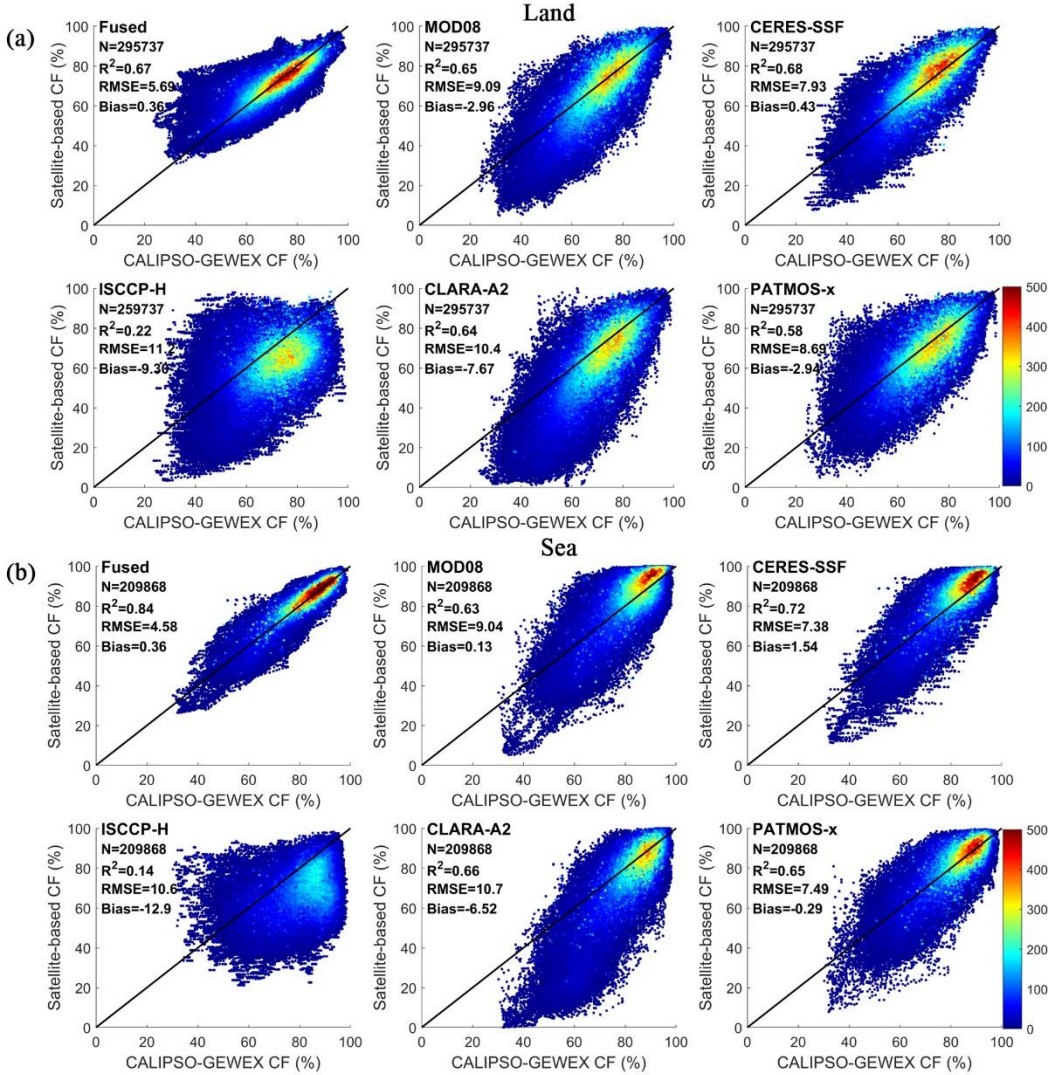

Figure 11. Validation of the fused cloud fraction and the original passive sensor datasets against the CALIPSO-GEWEX dataset over (a) land and (b) sea regions, with a temporal range from 2006 to 2016.

As the accepted reference for passive sensor products, CALIPSO-based products are
considered to provide excellent data and are always used to validate the accuracy of cloud datasets.
In Fig. 11, we compare the CFs of passive sensor products before and after fusion with that of the
CALIPSO-GEWEX product. The results show that when compared with the original satellite data,
the consistencies between the fused product and the active sensor product were further improved in
both land and sea regions. The RMSEs were reduced to about one-third to one-half of the original
values, or approximately 5.69% and 4.58% for land and sea regions, respectively. Actually, the
consistency of CFs between passive and active sensor datasets was higher than that between satellite
data and ground observations. Except for the ISCCP-H products, $R^2$ of original satellite data was
over 0.63; that of fused CF only improved obviously in sea regions (about 0.12–0.21), while it
improved slightly but in inconspicuously in land regions (about −0.01–0.1). This can be explained
by the fact that the fusion algorithm greatly improves the low-value CFs in the land areas (especially
on the GrIS) to levels similar to that of ground-based observations, while the CF of the active sensor
data was no more than 60%. Therefore, some overestimations for the fused CF existed when

compared with the CALIPSO-GEWEX CF data. From the bias of Fig. 11 (a), we also see that the fusion algorithm can obviously improve the CF underestimated by the original satellite data. However, in the sea regions, the MODIS-based datasets seem to overidentify CF, especially when the CF was over 80%. Meanwhile, the AVHRR-based datasets show underestimation when CF was less than 80%. Obviously, the fused product corrected these CFs to a more suitable range.

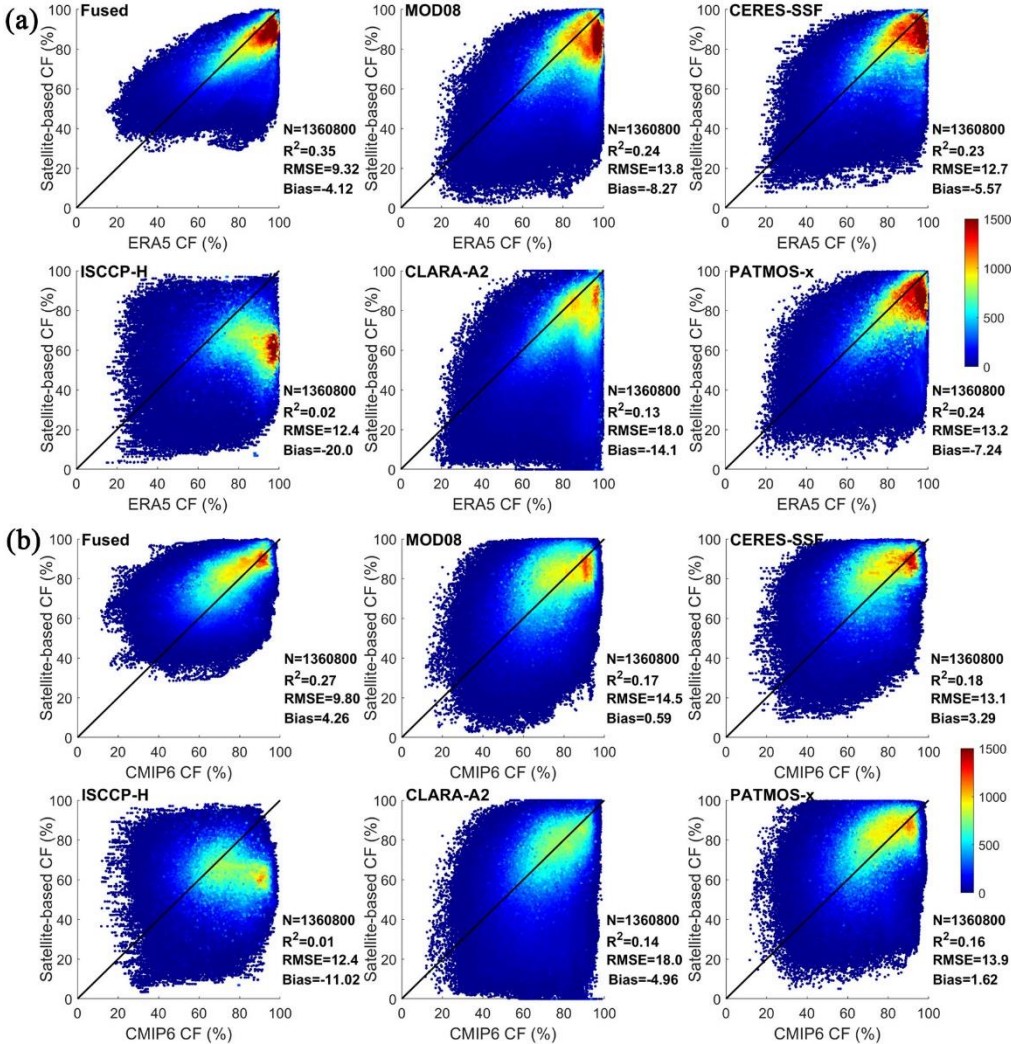

Figure 12. Validation of the fused cloud fraction (CF) and the original passive sensor datasets against (a) ERA5 CF dataset and (b) CMIP6 CF dataset over the Holarctic.

Reanalysis data and the climate model data are commonly used to provide a consistent and continuous dataset for long-term climate trends and variability studies. These datasets can provide insights into the behavior of the climate system that would be difficult to obtain from direct observations alone. To further show the advantages of the fusion results, we analyzed the difference in CFs between different satellite data, ERA5 reanalysis datasets and the MRI-AGCM3-2-S climate model. As can be seen from Fig. 12, the fusion product greatly reduced the deviation in CF between the satellite data and the reanalysis dataset and the model data. When compared with the ERA5 CF dataset, the scatters of fused CFs were more concentrated around the 1:1 line than those of the original satellite data. $R^2$ of the fusion product was about 1.5 times higher (improved about 0.18) than that of the original data, and the RMSEs and bias decreased to one-third of their original values (decreases of about 3.08–8.68% and 1.45–15.88%, respectively). This means that the distributions of the CFs over the entire Arctic of the fusion

product were more consistent with those of the reanalysis CF dataset than the original satellite. However, the low absolute values also indicated that there were inescapable inconsistencies in some grids. The ERA5 dataset has usually been reported to overestimate CF in some regions of the Arctic, especially in the ocean regions (Yeo et al., 2022). In these regions the fused CF has lightly higher values than that of the ERA5 data.

The comparison results with the MRI-AGCM3-2-S CF show that when compared with the original satellite data, the fusion method reduced the CF underestimation partly; these underestimations were often seen in April or over the central and western GrIS. In addition, $R^2$ was improved by about 0.14, and the RMSEs were reduced to one-fourth of their values of original satellite data (about 2.60–8.20% reduction). However, although the fusion data relieve some CF overestimations that occurred in original passive sensor datasets, the scatter plot in Fig. 12 shows that the fusion CFs in some grids were significantly higher than the CF of model data (with bias by 4.26%). These grids are usually found in the open water areas of Arctic Ocean, central Alaska, central Eurasia, and along the eastern margin of Greenland. Several studies have shown that the climate models underestimate the CF over these regions (Vignesh et al., 2020).

**5 Discussion**

**5.1 The Efficacy of CDF Matching in CF Fusion**

The CDF matching approach was operated based on a time series CF considering the time-varying process of CF products at a specific longitude–latitude grid box. Compared with the metrics for the traditional approach, the CF of multiple passive sensor products was scaled to a level similar to the active sensor CF after CDF matching, so that the inconsistencies among multiple passive sensor CF datasets were reduced. To further evaluate the efficacy of CDF matching in the CF fusion process, we quantitatively evaluated the deviation between satellite data before and after CDF correction with ground observation data.

By comparing Fig. 10 and Fig. 13, we can infer that CDF matching can obviously improve the low value of CFs typical of satellite data, making such data more similar to that observed by ground-based sites. These improvements were more obvious for CFs over land regions. Among them, the largest bias correction was seen for the ISCCP-H products (about 7.9% improvement) and the CLARA-A2 products (about 6.5% improvement); the former always underestimated CF in the Arctic (Kotarba, 2015; Liu et al., 2022) and the latter have often been reported to under-identify CF over northern Canada, northern Russia, and across the entire GrIS in land regions and over the entire Arctic Ocean in April (Karlsson and Dybbroe, 2010). Note that the bias of CERES-SSF changes from 0.4% to −0.72% after CDF matching, because CERES-SSF products are usually reported to overestimate CF and these overestimations were corrected reasonably. For the ocean regions, the ground references used in this paper were derived from multiple platform observations, which have great spatio-temporal heterogeneity. Therefore, a large CF discrepancy existed between satellite data and ocean observations. Almost all the passive sensor data have RMSEs and bias that would decrease after CDF correction by about 0.8–1.7% and 0.68–5.26%, respectively. The CDF matching mainly improves the CF in the high-value grid boxes of MODIS-based data and PATMOS-x data as well as in the CF in low-value grid boxes of ISCCP-H and CLARA-A2. Satellite observation covering open sea areas typically presents a higher CF compared to station observation. Consequently, partial overestimation may persist despite correction by the CDF

matching approach. In the subsequent fusion process, the difference between satellite CF and ground CF
was taken into account, which can play a certain extent role in overfitting correction.

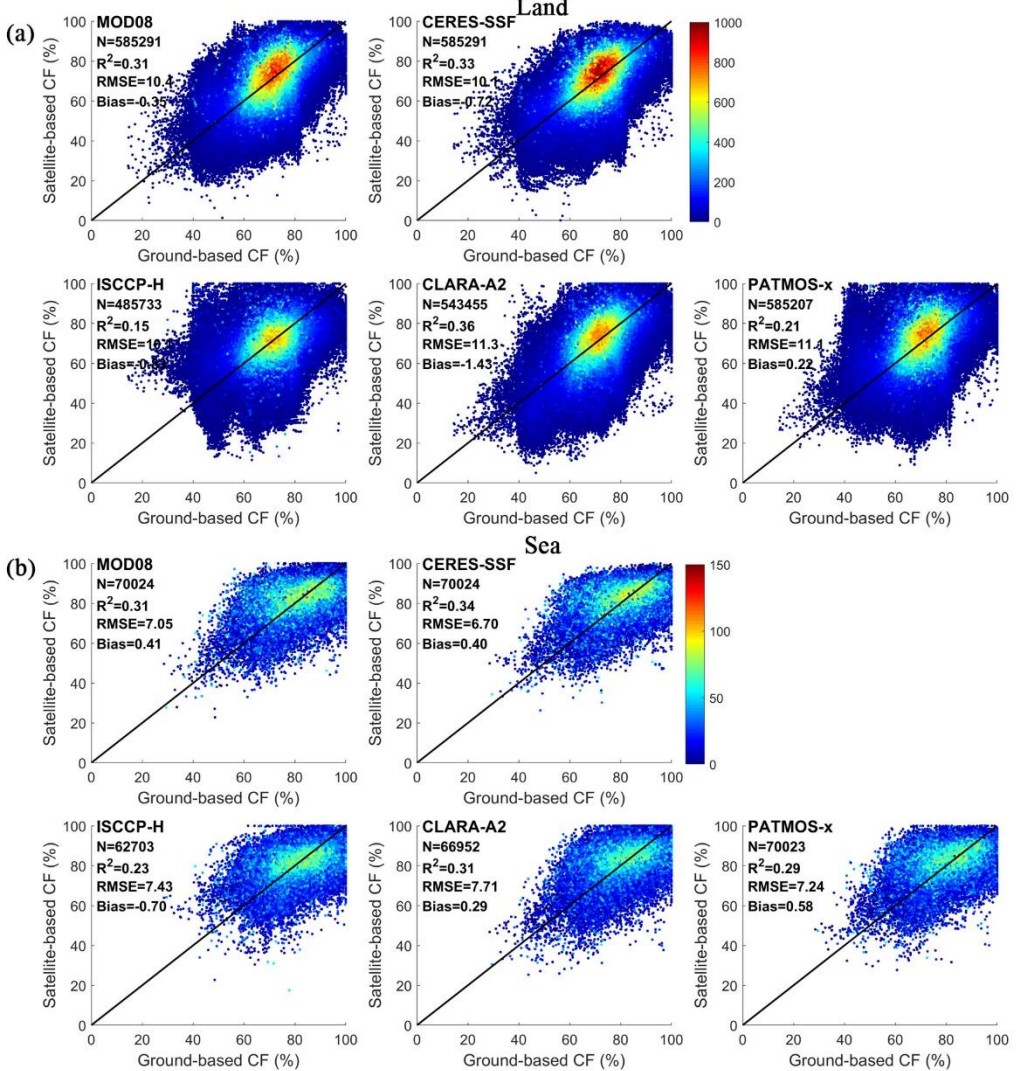


Figure 13. Validation of the corrected cloud fraction of passive sensor datasets after cumulative distribution
function matching against (a) CRU TS4.05 dataset over land regions and against (b) ICOADS dataset over sea
regions.

In addition, in the land area, CDF matching was directly carried out grid by grid. However, the short
temporal range (2006–2016) of the reference data limits the production of long time-series CF products.
In this study, we proposed a hypothesis that the matching parameter in a specific grid box does not change
over time. To prove the validity of this hypothesis, we conducted sensitivity analysis on the matching
parameters from the fifth to the eleventh year at one-year intervals. The findings indicate that any
deviations in matching parameters were under 0.05% when the time horizon exceeded 8 years. This
demonstrates a level of stability in the correction coefficient when utilizing data for a period exceeding
11 years (Figure A1). Figure 14 displays the variation in differences between satellite data and ground
observations before and after conducting CDF matching throughout the duration of the study. These
differences are calculated by subtracting the deviation between satellite data and ground observations
subsequent to CDF matching from that prior to CDF matching. Clearly, the differences remained steady
over time, and the maximum average annual difference was no more than 1.56%, while part of it was
derived from the orbit drift of satellite and variations in the spectral channel.

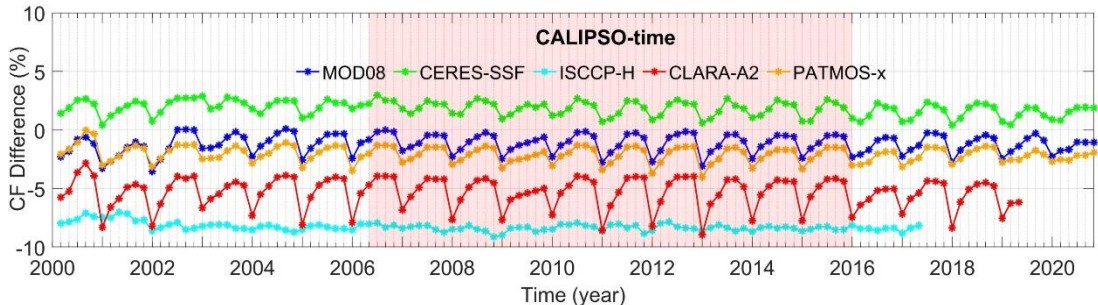

Figure 14. The difference in results between satellite data and ground observations before and after cumulative
distribution function matching over the Arctic from 2000 to 2020.

**5.2 The Uncertainties of the original Satellite Data**

CF products from different sensors have different degree of uncertainties. As a knowledge-centered
approach, Bayesian maximum entropy approach could integrate informative content with uncertainty
from different sources based on a rigorous theoretical support of considerable generality to achieve
improved prediction accuracy. For example, the observed data that accompanied by obvious sources of
uncertainty such as inaccuracy in measuring devices, modeling uncertainties, and human error are treated
as soft data in BME strategy. For the CF datasets of multiple satellite, the uncertainties come from
calibration error, orbit drift, signal degradation as well as the errors of cloud detection algorithms (Liu et
al., 2022). To achieve optimum estimation of CFs by combining data from multiple sensors, it is
imperative to explicitly consider the uncertainties associated with the CF data that is being merged. In
our study, the CF data of passive sensor products are viewed as soft data, and the uncertainty associated
with different error sources can be expressed explicitly by probability distribution.
Specifically, the soft data of multiple satellite CF datasets were constructed by comparing the
spatiotemporally collocated satellite CFs and the ground-based records from CRU TS4.05 over land and
from ICOADS over sea. Traditionally, the deviations between each satellite dataset and ground site
observations at different times and different regions have been averaged to the entire datasets, and then
used to calculate the average uncertainty of these data. In this way, the spatial variation of uncertainty in
each satellite dataset was ignored. Because the conditions that cause uncertainty are variable in time and
space, the uncertainties in each satellite dataset were definitely not the same everywhere (Tang et al.,
2016). This is especially true for the ICOADS data, which come from different platforms and introduce
large inconsistencies in results. In this study, we constructed soft data for CF over land, ocean, and GrIS
regions every month separately by analyzing the PDF differences for different regions and different
months, which realized more consistent results with the ground observations. However, despite concerted
efforts, determining the uncertainty for each grid remains challenging in light of the substantial temporal
and spatial gaps of the reference data, particularly that which pertains to the marine domain.
**5.3 The Uncertainties of the Fusion CF**
To assess the fusion algorithm's reliability, we used the standard deviation of error within each grid value
in the fusion process to quantify the uncertainties. Specifically, we determined the standard deviation of
the predicted posterior probability density function on each grid point. Our findings demonstrate that,

with the exception of the northern region of Greenland and part of the margin error, the standard deviation of error in other areas was within 3% (Fig 15). We attribute these discrepancies primarily to the underestimation of satellite observations, particularly the ISCCP-H data, by around 10-30% in the central zone of Greenland. Moreover, the CF of ISCCP-H was significantly overestimated beyond the Greenland margin. Such significant inconsistencies can adversely affect the fusion results. In addition, because the CF of satellite data, particularly satellite data based on AVHRR, was significantly lower than that of ground observation data and active sensor data in April, and a significant difference existed between different datasets, the standard deviation of error after fusion marginally increased in April, with some areas at approximately 4%. It should be noted that our fused data shows an overestimation in the Greenland region. This is mainly because the fusion process prioritizes consistency between the fused data and ground observations. In specific applications, users can make corresponding adjustments based on active sensor data for calibration purposes.

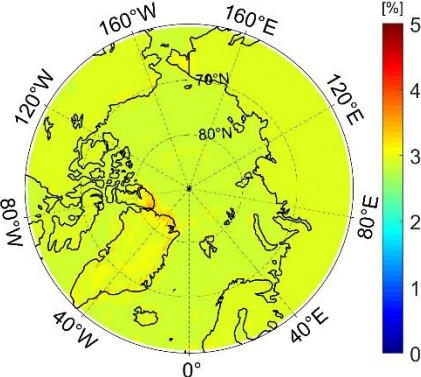

Figure 15. The mean error standard deviation of the fusion results

**6 Data Availability**

The fused CF product is available on the Zenodo repository at https://doi.org/10.5281/zenodo.7624605 (Liu et al., 2022). The gridded CF data are provided both in *.mat format (Fused_CF_Arctic_MAT, with file size 9.91 MB) and netCDF format (Fused_CF_Arctic_netCDF, with file size 10.7MB) at 1° spatial resolution and monthly temporal resolution during 2000–2020 in percentages. The results in these two folders are exactly the same, someone can download either format as needed.

**7 Conclusions**

The spatiotemporal inconsistency in existing satellite CF products would inhibit their application in climatological and energy budget studies. Over the Arctic region, the special climatic conditions and underlying surface characteristics limit the cloud detection abilities of passive/optical satellite sensors. The complementary features of the CF products derived from multiple satellite sensors in spatial completeness and accuracy make it possible to produce an improved CF product by merging data from multi-sensor satellite CF products.

In this paper, we propose a data fusion strategy for producing high-quality monthly CF data over the entire Arctic with a latitude larger than 60°N during sunlit months from 2000 to 2020. Four key steps were involved in the proposed strategy: (1) data quality control; (2) correct the bias of passive sensor

data using CDF matching; (3) obtain the spatiotemporally isotropous component by removing the spatiotemporal trends; and (4) produce very accurate CF data by fusing multiple satellite products and considering the uncertainty between satellite data and ground observations with the BME approach.

The fusion algorithm proposed by this study apparently reduced inconsistencies in the Arctic CF data acquired by multiple satellite products and the reference products spatiotemporally, resulting in 10–20% reductions of CF differences between fused satellite products and the reference data, and an obvious improvement was seen across the GrIS and in the central Arctic Ocean. The results from 21-year data sets in the study areas demonstrate that the monthly mean CF of the fusion product varied from 62% (April) to 79% (September) during the study period, which is similar to that of the two reference datasets. After CDF matching, the inconsistencies of multiple satellite CF products were reduced by about 3.43% for the entire Arctic, with a larger reduction (4.46%) for sea ice regions. The overestimation of MODIS-based products and the underestimation of AVHRR-based products have been effectively corrected, with the CERES-SSF bias changing from 0.4% to −0.72% and the bias of ISCCP-H and CLARA-A2 decreasing by about 7.9% and 6.5%, respectively. After BME fusing, comparisons with the ground-based observations (CRU TS4.05 in land and ICOADS in marine areas) and the active sensor data CALIPSO-GEWEX show that $R^2$ improved by about 0.05–0.48 for different products; meanwhile, the overall RMSEs and bias of fusion product were reduced by about 2.08–7.75% and 1.6–12.54%, with reductions of nearly 50% and 67% when compared with that of the original passive sensor data, respectively. When compared with the reanalysis CF dataset ERA5 and the model dataset MRI-AGCM3-2-S, $R^2$ increased by about 0.18 and 0.14, RMSE and bias for reanalysis data decreased by about one-third of that for the original data, with reductions about 3.08–8.68% and 1.45–15.88% for different data, respectively. The RMSEs for model data dropped to one-fourth of their original values (about a 2.60–8.20% reduction). These mean that the proposed fusion algorithm effectively removed CF data with differences greater than 30% and made the fused Arctic CF estimation more robust than those data from a single satellite. Nevertheless, the fused product could completely cover the entire Arctic, especially the ocean regions, where the active sensor data and the ground-based data have large data gaps. Temporally, the fused data can complement the missing data caused by the faults of satellites carrying AVHRR sensors and the absence of Aqua data before 2002 as well as the temporal limitation of passive sensors.

In general, the proposed fusion algorithm combines the complementary features of multiple satellite CF datasets; it not only takes full advantage of the spatiotemporal autocorrelation among neighboring grids but also incorporates uncertainty estimates of multi-sensor CFs, such as the uncertainties of each passive sensor dataset, the uncertainties between passive and active sensor datasets, as well as the uncertainties between satellite data and ground-based observations. Through temporal and spatial expansion schemes, this fusion framework makes up for the disadvantages in spatiotemporal ranges of reference data. Finally, the fusion algorithm can generate monthly $1° \times 1°$ CF product covering the entire Arctic region during 2000 to 2020, which has positive significance for reducing the uncertainties of assessment of surface radiation flux and improving the accuracy of research related to climate change and energy budgets both regionally and globally. However, some overestimations were observed, especially in ocean regions. This may be attributed to the fact that the ocean stations are too sparse to play a certain role in correcting the overfitting of CDF. Although ICOADS is a widely used ocean validation dataset, it has great spatiotemporal heterogeneity because it comes from a variety of different observation platforms and the sampling is affected by the extent of sea ice. Better reference data should be explored to further improve the uncertainty involved in the assessment of the fused product.

**Appendix A**

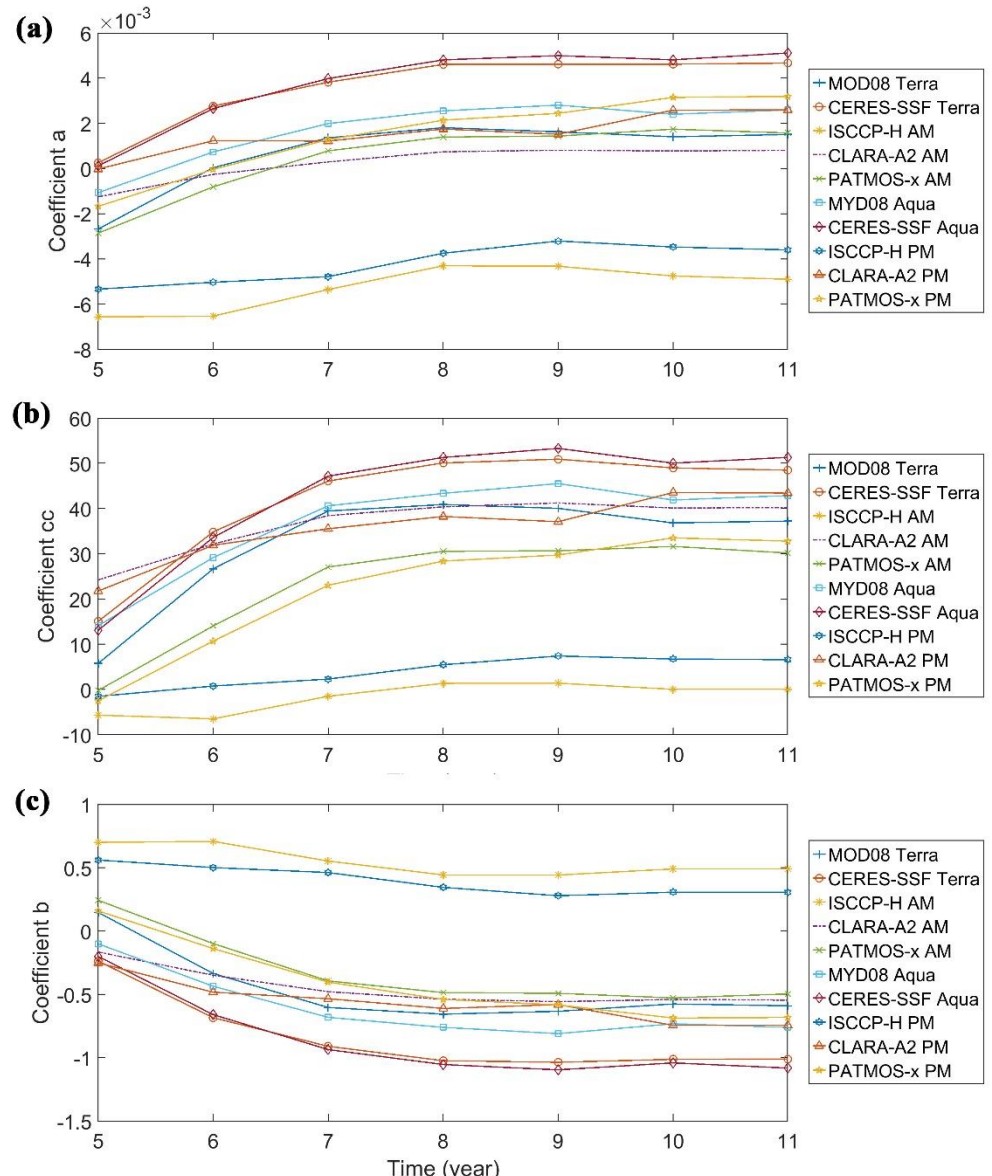


Figure A1. The sensitivity analysis on the CDF matching parameters from the fifth (2011) to the eleventh year (2017)
of CALIPSO time at one-year intervals. The Coefficient a, b and c are calculated by the least-square fit method. And
the time period only contains sunlight month from April to September.
**Author contributions**
XL performed the method, validation, and writing the original draft of the paper. TH was responsible
for conceptualization, supported and supervised the study and reviewed the paper. SL was responsible
for conceptualization and reviewed the paper. RL provided guidance on data processing. XX, RM and
YM contributed to the editing and revising of the manuscript. XL prepared the manuscript with
contributions from all co-authors.

**831**    **Competing interests**

**832**    The contact author has declared that none of the authors has any competing interests.

**833**    **Disclaimer**

**834**    Publisher's note: Copernicus Publications remains neutral with regard to jurisdictional claims in
**835**    published maps and institutional affiliations.

**836**    **Acknowledgments**

**837**    We thank the relevant teams and organizations for providing the data sets used in this study. We
**838**    thank the NASA Level-1 and Atmosphere Archive & Distribution System Distributed Active Archive
**839**    Center (LAADS DAAC) for providing the MOD08_M3/MYD08_M3 products, the NASA Langley
**840**    Research Center Atmospheric Science Data Center (ASDC) for providing CERES_SSF and CALIPSO-
**841**    GEWEX data, the Satellite Application Facility on Climate Monitoring (CM SAF) for providing the
**842**    CLARA-A2 product, the NOAA National Centers for Environmental Information (NCEI) for providing
**843**    PATMOS-x and ISCCP-H products. We also thank the University of East Anglia Climatic Research
**844**    Unit for their providing the CRU TS4.05 data, the NOAA Physical Sciences Laboratory for their
**845**    providing the ICOADS marine data, and the European Centre for Medium-Range Weather Forecasts
**846**    (ECMWF) for their archiving the ERA5 data. Many thanks to the LetPub (http://letpub.com.cn/) for its
**847**    linguistic assistance during the preparation of this manuscript.

**848**    **Financial support**

**849**    This work was supported by the National Natural Science Foundation of China Grant (42090012),
**850**    Hubei Natural Science Foundation Grant (2021CFA082) and National Key Research and Development
**851**    Program of China (2020YFA0608704).

**852**    **References**

**853**    Ackerman, S. A., Holz, R. E., Frey, R., Eloranta, E. W., Maddux, B. C., and McGill, M.: Cloud detection
**854**    with MODIS. Part II: Validation, Journal of Atmospheric and Oceanic Technology, 25, 1073-1086,
**855**    10.1175/2007jtecha1053.1, 2008.

**856**    Beckerman, B. S., Jerrett, M., Serre, M., Martin, R. V., Lee, S.-J., van Donkelaar, A., Ross, Z., Su, J., and
**857**    Burnett, R. T.: A Hybrid Approach to Estimating National Scale Spatiotemporal Variability of PM2.5 in
**858**    the Contiguous United States, Environmental Science & Technology, 47, 7233-7241, 10.1021/es400039u,
**859**    2013.

**860**    Bogaert, P., Christakos, G., Jerrett, M., and Yu, H. L.: Spatiotemporal modelling of ozone distribution in
**861**    the State of California, Atmospheric Environment, 43, 2471-2480, 10.1016/j.atmosenv.2009.01.049,
**862**    2009.

Bojinski, S., Verstraete, M., Peterson, T. C., Richter, C., Simmons, A., and Zemp, M.: THE CONCEPT
OF ESSENTIAL CLIMATE VARIABLES IN SUPPORT OF CLIMATE RESEARCH, APPLICATIONS,
AND POLICY, B Am Meteorol Soc, 95, 1431-1443, 10.1175/bams-d-13-00047.1, 2014.
Brocca, L., Hasenauer, S., Lacava, T., Melone, F., Moramarco, T., Wagner, W., Dorigo, W., Matgen, P.,
Martínez-Fernández, J., Llorens, P., Latron, J., Martin, C., and Bittelli, M.: Soil moisture estimation
through ASCAT and AMSR-E sensors: An intercomparison and validation study across Europe, Remote
Sensing of Environment, 115, 3390-3408, 10.1016/j.rse.2011.08.003, 2011.
Chatterjee, A., Michalak, A. M., Kahn, R. A., Paradise, S. R., Braverman, A. J., and Miller, C. E.: A
geostatistical data fusion technique for merging remote sensing and ground-based observations of aerosol
optical thickness, Journal of Geophysical Research-Atmospheres, 115, 10.1029/2009jd013765, 2010.
Christakos, G.: Modern Spatiotemporal Geostatistics, Modern spatiotemporal geostatistics2000.
Christakos, G.: Integrative problem-solving in a time of decadence, Springer Science & Business Media
875 2010.

Christakos, G., Kolovos, A., Serre, M. L., and Vukovich, F.: Total ozone mapping by integrating
databases from remote sensing instruments and empirical models, Ieee Transactions on Geoscience and
Remote Sensing, 42, 991-1008, 10.1109/Tgrs.2003.822751, 2004.
Christakos G, Serre ML.: BME analysis of spatiotemporal particulate matter distributions in North
Carolina. Atmospheric Environment 34:3393–3406,2000.
Claudia, S., William R., Stefan K.: Assessment of Global Cloud Data Sets from Satellites A Project of
the World Climate Research Programme Global Energy and Water Cycle Experiment (GEWEX)
Radiation Panel, World Climate Research Program Proport, 2012.
Cressie, N.: Statistics for spatial data, John Wiley & Sons2015.
Danso, D. K., Anquetin, S., Diedhiou, A., Kouadio, K., and Kobea, A. T.: Daytime low-level clouds in
West Africa - occurrence, associated drivers, and shortwave radiation attenuation, Earth Syst Dynam, 11,
1133-1152, 10.5194/esd-11-1133-2020, 2020.
Doelling, D. R., Sun, M., Nguyen, L. T., Nordeen, M. L., Haney, C. O., Keyes, D. F., and Mlynczak, P.
E.: Advances in Geostationary-Derived Longwave Fluxes for the CERES Synoptic (SYN1deg) Product,
Journal of Atmospheric and Oceanic Technology, 33, 503-521, 10.1175/Jtech-D-15-0147.1, 2016.
Drusch, M.: Observation operators for the direct assimilation of TRMM microwave imager retrieved soil
moisture, Geophysical Research Letters, 32, 10.1029/2005gl023623, 2005.
Eastman, R. and Warren, S. G.: Arctic Cloud Changes from Surface and Satellite Observations, Journal
of Climate, 23, 4233-4242, 10.1175/2010jcli3544.1, 2010.
Enriquez-Alonso, A., Sanchez-Lorenzo, A., Calbo, J., Gonzalez, J. A., and Norris, J. R.: Cloud cover
climatologies in the Mediterranean obtained from satellites, surface observations, reanalyses, and CMIP5
simulations: validation and future scenarios, Climate Dynamics, 47, 249-269, 10.1007/s00382-015-
898 2834-4, 2016.

Forbes, R. M. and Ahlgrimm, M.: On the Representation of High-Latitude Boundary Layer Mixed-Phase
Cloud in the ECMWF Global Model %J Monthly Weather Review, 142, 3425-3445,
https://doi.org/10.1175/MWR-D-13-00325.1, 2014.

Freeman, E., Woodruff, S. D., Worley, S. J., Lubker, S. J., Kent, E. C., Angel, W. E., Berry, D. I., Brohan, P., Eastman, R., Gates, L., Gloeden, W., Ji, Z., Lawrimore, J., Rayner, N. A., Rosenhagen, G., and Smith, S. R.: ICOADS Release 3.0: a major update to the historical marine climate record, International Journal of Climatology, 37, 2211-2232, 10.1002/joc.4775, 2017.

Fuentes, M. and Raftery, A. E.: Model evaluation and spatial interpolation by Bayesian combination of observations with outputs from numerical models, Biometrics, 61, 36-45, 10.1111/j.0006-341X.2005.030821.x, 2005.

Gao, F., Masek, J., Schwaller, M., Hall, F. J. I. T. o. G., and sensing, R.: On the blending of the Landsat and MODIS surface reflectance: Predicting daily Landsat surface reflectance, 44, 2207-2218, 2006.

Griffith, D. A.: STATISTICS FOR SPATIAL DATA - CRESSIE,N, Geographical Analysis, 25, 271-275, 1993.

Hakuba, M. Z., Folini, D., Wild, M., Long, C. N., Schaepman-Strub, G., and Stephens, G. L.: Cloud effects on atmospheric solar absorption in light of most recent surface and satellite measurements, 10.1063/1.4975543, 2017.

Harris, I., Jones, P. D., Osborn, T. J., and Lister, D. H.: Updated high-resolution grids of monthly climatic observations - the CRU TS3.10 Dataset, International Journal of Climatology, 34, 623-642, 10.1002/joc.3711, 2014.

Harris, I., Osborn, T. J., Jones, P., and Lister, D.: Version 4 of the CRU TS monthly high-resolution gridded multivariate climate dataset, Sci Data, 7, 109, 10.1038/s41597-020-0453-3, 2020.

He, J. and Kolovos, A.: Bayesian maximum entropy approach and its applications: a review, Stochastic Environmental Research and Risk Assessment, 32, 859-877, 10.1007/s00477-017-1419-7, 2017.

Heidinger, A. K., Evan, A. T., Foster, M. J., and Walther, A.: A Naive Bayesian Cloud-Detection Scheme Derived from CALIPSO and Applied within PATMOS-x, Journal of Applied Meteorology and Climatology, 51, 1129-1144, 10.1175/Jamc-D-11-02.1, 2012.

Heidinger, A. K., Foster, M. J., Walther, A., and Zhao, X. P.: The Pathfinder Atmospheres-Extended Avhrr Climate Dataset, Bulletin of the American Meteorological Society, 95, 909-+, 10.1175/Bams-D-12-00246.1, 2014.

Hilker, T., Wulder, M. A., Coops, N. C., Linke, J., McDermid, G., Masek, J. G., Gao, F., and White, J. C.: A new data fusion model for high spatial- and temporal-resolution mapping of forest disturbance based on Landsat and MODIS, Remote Sensing of Environment, 113, 1613-1627, 10.1016/j.rse.2009.03.007, 2009.

Hollmann, R.: ESA Cloud_cci Product Validation and Intercomparison Report(PVIR), 10.5676/DWD/ESA_Cloud_cci/AVHRR-PM/V002, 2018.

Hollmann, R., Merchant, C. J., Saunders, R., Downy, C., Buchwitz, M., Cazenave, A., Chuvieco, E., Defourny, P., de Leeuw, G., Forsberg, R., Holzer-Popp, T., Paul, F., Sandven, S., Sathyendranath, S., van Roozendael, M., and Wagner, W.: THE ESA CLIMATE CHANGE INITIATIVE Satellite Data Records for Essential Climate Variables, B Am Meteorol Soc, 94, 1541-1552, 10.1175/bams-d-11-00254.1, 2013.

Hu, M. and Xue, M. J. G. r. l.: Implementation and evaluation of cloud analysis with WSR-88D reflectivity data for GSI and WRF-ARW, 34, 2007.

Huang, Y. Y., Dong, X. Q., Xi, B. K., Dolinar, E. K., Stanfield, R. E., and Qiu, S. Y.: Quantifying the Uncertainties of Reanalyzed Arctic Cloud and Radiation Properties Using Satellite Surface Observations, Journal of Climate, 30, 8007-8029, 10.1175/Jcli-D-16-0722.1, 2017.

Hunt, W. H., Winker, D. M., Vaughan, M. A., Powell, K. A., Lucker, P. L., and Weimer, C.: CALIPSO Lidar Description and Performance Assessment, Journal of Atmospheric and Oceanic Technology, 26, 1214-1228, 10.1175/2009jtecha1223.1, 2009.

Jaynes, E. T. J. P. r.: Information theory and statistical mechanics, 106, 620, 1957.

Jin, W., Fu, R.-d., Ye, M., and Li, J.-x.: Meteorological Cloud Image Fusion Using Contourlet Transform and Compressed Sensing, International Conference on Ecological Protection of Lakes-Wetlands-Watershed and Application of 3S Technology (EPLWW3S 2011), Nanchang, PEOPLES R CHINA, 2011

Jun 25-26, WOS:000391516000097, 413-416, 2011.

Karlsson, K.-G., Anttila, K., Trentmann, J., Stengel, M., Meirink, J. F., Devasthale, A., Hanschmann, T., Kothe, S., Jaaskelainen, E., Sedlar, J., Benas, N., van Zadelhoff, G.-J., Schlundt, C., Stein, D., Finkensieper, S., Hakansson, N., and Hollmann, R.: CLARA-A2: the second edition of the CM SAF cloud and radiation data record from 34 years of global AVHRR data, Atmospheric Chemistry and Physics, 17, 5809-5828, 10.5194/acp-17-5809-2017, 2017.

Karlsson, K. G. and Devasthale, A.: Inter-Comparison and Evaluation of the Four Longest Satellite-Derived Cloud Climate Data Records: CLARA-A2, ESA Cloud CCI V3, ISCCP-HGM, and PATMOS-x, Remote Sens-Basel, 10, 10.3390/rs10101567, 2018.

Karlsson, K. G. and Dybbroe, A.: Evaluation of Arctic cloud products from the EUMETSAT Climate Monitoring Satellite Application Facility based on CALIPSO-CALIOP observations, Atmospheric Chemistry and Physics, 10, 1789-1807, DOI 10.5194/acp-10-1789-2010, 2010.

Karlsson, K. G. and Hakansson, N.: Characterization of AVHRR global cloud detection sensitivity based on CALIPSO-CALIOP cloud optical thickness information: demonstration of results based on the CM SAF CLARA-A2 climate data record, Atmos Meas Tech, 11, 633-649, 10.5194/amt-11-633-2018, 2018.

Karlsson, K. G., Riihela, A., Mueller, R., Meirink, J. F., Sedlar, J., Stengel, M., Lockhoff, M., Trentmann, J., Kaspar, F., Hollmann, R., and Wolters, E.: CLARA-A1: a cloud, albedo, and radiation dataset from 28 yr of global AVHRR data, Atmospheric Chemistry and Physics, 13, 5351-5367, 10.5194/acp-13-5351-2013, 2013.

Kato, S., Loeb, N. G., Rutan, D. A., Rose, F. G., Sun-Mack, S., Miller, W. F., and Chen, Y.: Uncertainty Estimate of Surface Irradiances Computed with MODIS-, CALIPSO-, and CloudSat-Derived Cloud and Aerosol Properties, Surveys in Geophysics, 33, 395-412, 10.1007/s10712-012-9179-x, 2012.

Kato, S., Rose, F. G., Rutan, D. A., Thorsen, T. J., Loeb, N. G., Doelling, D. R., Huang, X., Smith, W. L., Su, W., and Ham, S.-H.: Surface Irradiances of Edition 4.0 Clouds and the Earth's Radiant Energy System (CERES) Energy Balanced and Filled (EBAF) Data Product, J Climate, 31, 4501-4527, 10.1175/jcli-d-17-0523.1, 2018a.

Kato, S., Rose, F. G., Rutan, D. A., Thorsen, T. J., Loeb, N. G., Doelling, D. R., Huang, X. L., Smith, W. L., Su, W. Y., and Ham, S. H.: Surface Irradiances of Edition 4.0 Clouds and the Earth's Radiant Energy System (CERES) Energy Balanced and Filled (EBAF) Data Product, Journal of Climate, 31, 4501-4527, 10.1175/Jcli-D-17-0523.1, 2018b.

Kato, S., Rose, F. G., Sun-Mack, S., Miller, W. F., Chen, Y., Rutan, D. A., Stephens, G. L., Loeb, N. G., Minnis, P., Wielicki, B. A., Winker, D. M., Charlock, T. P., Stackhouse, P. W., Xu, K.-M., and Collins, W. D.: Improvements of top-of-atmosphere and surface irradiance computations with CALIPSO-, CloudSat-, and MODIS-derived cloud and aerosol properties, Journal of Geophysical Research, 116, 10.1029/2011jd016050, 2011.

Kennedy, A., Xi, B., Dong, X., and Zib, B. J.: Evaluation and Intercomparison of Cloud Fraction and Radiative Fluxes in Recent Reanalyses over the Arctic Using BSRN Surface Observations, J Climate, 25, 2291-2305, 10.1175/jcli-d-11-00147.1, 2012.

Kenyon, J. S., Moninger, W. R., Smith, T. L., Peckham, S. E., Lin, H., Grell, G. A., Dowell, D. C., James, E. P., Olson, J. B., Smirnova, T. G., Alexander, C. R., Hu, M., Brown, J. M., Weygandt, S. S., Benjamin, S. G., and Manikin, G. S.: A North American Hourly Assimilation and Model Forecast Cycle: The Rapid Refresh, Mon Weather Rev, 144, 1669-1694, 10.1175/mwr-d-15-0242.1, 2016.

Kim, D. and Ramanathan, V.: Solar radiation budget and radiative forcing due to aerosols and clouds, Journal of Geophysical Research, 113, 10.1029/2007jd008434, 2008.

Kotarba, A. Z.: Evaluation of ISCCP cloud amount with MODIS observations, Atmos Res, 153, 310-317, 10.1016/j.atmosres.2014.09.006, 2015.

Kotarba, A. Z.: Calibration of global MODIS cloud amount using CALIOP cloud profiles, Atmospheric Measurement Techniques, 13, 4995-5012, 10.5194/amt-13-4995-2020, 2020.

Li, A., Bo, Y., Zhu, Y., Guo, P., Bi, J., and He, Y.: Blending multi-resolution satellite sea surface temperature (SST) products using Bayesian maximum entropy method, Remote Sensing of Environment, 135, 52-63, 10.1016/j.rse.2013.03.021, 2013.

Li, L., Shi, R., Zhang, L., Zhang, J., and Gao, W.: The data fusion of aerosol optical thickness using universal kriging and stepwise regression in East China, Conference on Remote Sensing and Modeling of Ecosystems for Sustainability XI, San Diego, CA, 2014

Aug 18-20, WOS:000344548600027,    10.1117/12.2061764, 2014.

Li, S. and Yang, B.: Multifocus image fusion by combining curvelet and wavelet transform, Pattern Recognition Letters, 29, 1295-1301, 10.1016/j.patrec.2008.02.002, 2008.

Liu, X., He, T., Sun, L., Xiao, X., Liang, S., and Li, S.: Analysis of Daytime Cloud Fraction Spatiotemporal Variation over the Arctic from 2000 to 2019 from Multiple Satellite Products, Journal of Climate, 35, 3995-4023, 10.1175/jcli-d-22-0007.1, 2022.

Liu, Y., Liu, S., and Wang, Z.: A general framework for image fusion based on multi-scale transform and sparse representation, Information Fusion, 24, 147-164, 10.1016/j.inffus.2014.09.004, 2015.

Liu, Y., Wu, W., Jensen, M. P., and Toto, T.: Relationship between cloud radiative forcing, cloud fraction and cloud albedo, and new surface-based approach for determining cloud albedo, Atmospheric Chemistry and Physics, 11, 7155-7170, 10.5194/acp-11-7155-2011, 2011a.

Liu, Y., Ackerman, S. A., Maddux, B. C., Key, J. R., and Frey, R. A.: Errors in Cloud Detection over the Arctic Using a Satellite Imager and Implications for Observing Feedback Mechanisms, Journal of Climate, 23, 1894-1907, 10.1175/2009jcli3386.1, 2010.

Liu, Y., Key, J. R., Liu, Z., Wang, X., and Vavrus, S. J.: A cloudier Arctic expected with diminishing sea

ice, Geophysical Research Letters, 39, n/a-n/a, 10.1029/2012gl051251, 2012a.

Liu, Y. H., Key, J. R., Ackerman, S. A., Mace, G. G., and Zhang, Q. Q.: Arctic cloud macrophysical characteristics from CloudSat and CALIPSO, Remote Sensing of Environment, 124, 159-173, 10.1016/j.rse.2012.05.006, 2012b.

Liu, Y. Y., Parinussa, R. M., Dorigo, W. A., De Jeu, R. A. M., Wagner, W., van Dijk, A. I. J. M., McCabe, M. F., and Evans, J. P.: Developing an improved soil moisture dataset by blending passive and active microwave satellite-based retrievals, Hydrology and Earth System Sciences, 15, 425-436, 10.5194/hess-15-425-2011, 2011b.

Loyola R, D. G., Thomas, W., Spurr, R., and Mayer, B.: Global patterns in daytime cloud properties derived from GOME backscatter UV-VIS measurements, Int J Remote Sens, 31, 4295-4318, 10.1080/01431160903246741, 2010.

Marchant, B., Platnick, S., Meyer, K., and Wind, G.: Evaluation of the MODIS Collection 6 multilayer cloud detection algorithm through comparisons with CloudSat Cloud Profiling Radar and CALIPSO CALIOP products, Atmospheric Measurement Techniques, 13, 3263-3275, 10.5194/amt-13-3263-2020, 2020.

Marchant, B., Platnick, S., Meyer, K., Arnold, G. T., and Riedi, J.: MODIS Collection 6 shortwave-derived cloud phase classification algorithm and comparisons with CALIOP, Atmos Meas Tech, 9, 1587-1599, 10.5194/amt-9-1587-2016, 2016.

Miao, Q. and Wang, B.: A Novel Image Fusion Method Using Contourlet Transform, International Conference on Communications,

Minnis, P., Sun-Mack, S., Young, D. F., Heck, P. W., Garber, D. P., Chen, Y., Spangenberg, D. A., Arduini, R. F., Trepte, Q. Z., Smith, W. L., Ayers, J. K., Gibson, S. C., Miller, W. F., Hong, G., Chakrapani, V., Takano, Y., Liou, K. N., Xie, Y., and Yang, P.: CERES Edition-2 Cloud Property Retrievals Using TRMM VIRS and Terra and Aqua MODIS Data-Part I: Algorithms, Ieee Transactions on Geoscience and Remote Sensing, 49, 4374-4400, 10.1109/tgrs.2011.2144601, 2011.

Nazelle, A. D., Arunachalam, S., and Serre, M. L.: Bayesian maximum entropy integration of ozone observations and model predictions: an application for attainment demonstration in North Carolina, Environ Sci Technol, 44, 5707-5713, 10.1021/es100228w, 2010.

Nie, S., Wu, T., Luo, Y., Deng, X., Shi, X., Wang, Z., Liu, X., and Huang, J.: A strategy for merging objective estimates of global daily precipitation from gauge observations, satellite estimates, and numerical predictions, Advances in Atmospheric Sciences, 33, 889-904, 10.1007/s00376-016-5223-y, 2016.

Paul, A. H.: Collection 6.1 Change Summary Document MODIS Atmosphere Level-3 Algorithm and Global Products, 2017.

Philipp, D., Stengel, M., and Ahrens, B.: Analyzing the Arctic Feedback Mechanism between Sea Ice and Low-Level Clouds Using 34 Years of Satellite Observations, Journal of Climate, 33, 7479-7501, 10.1175/jcli-d-19-0895.1, 2020.

Poulsen, C. J., Tabor, C., and White, J.: Response to Comment on "Long-term climate forcing by atmospheric oxygen concentrations", Science, 353, 10.1126/science.aad8550, 2016.

Qian, Y., Long, C. N., Wang, H., Comstock, J. M., McFarlane, S. A., and Xie, S.: Evaluation of cloud fraction and its radiative effect simulated by IPCC AR4 global models against ARM surface observations, Atmospheric Chemistry and Physics, 12, 1785-1810, 10.5194/acp-12-1785-2012, 2012.

Ramanathan, V., Cess, R. D., Harrison, E. F., Minnis, P., Barkstrom, B. R., Ahmad, E., and Hartmann, D.: Cloud-Radiative Forcing and Climate - Results from the Earth Radiation Budget Experiment, Science, 243, 57-63, DOI 10.1126/science.243.4887.57, 1989.

Rossow, W. B. and Schiffer, R. A.: Advances in understanding clouds from ISCCP, Bulletin of the American Meteorological Society, 80, 2261-2287, 10.1175/1520-0477(1999)080<2261:Aiucfi>2.0.Co;2, 1999.

Savelyeva, E., Utkin, S., Kazakov, S., and Demyanov, V.: Modeling Spatial Uncertainty for Locally Uncertain Data, 7th International Conference on Geostatistics for Environmental Applications, Southampton, ENGLAND, 2010

Sep, WOS:000288481100026, 295-+, 10.1007/978-90-481-2322-3_26, 2010.

Shupe, M. D., Turner, D. D., Walden, V. P., Bennartz, R., Cadeddu, M. P., Castellani, B. B., Cox, C. J., Hudak, D. R., Kulie, M. S., Miller, N. B., Neely, R. R., Neff, W. D., and Rowe, P. M.: HIGH AND DRY New Observations of Tropospheric and Cloud Properties above the Greenland Ice Sheet, B Am Meteorol Soc, 94, 169-+, 10.1175/Bams-D-11-00249.1, 2013.

Sledd, A. and L'Ecuyer, T. S.: Emerging Trends in Arctic Solar Absorption, Geophys Res Lett, 48, 10.1029/2021gl095813, 2021.

Spadavecchia, L. and Williams, M.: Can spatio-temporal geostatistical methods improve high resolution regionalisation of meteorological variables?, Agricultural and Forest Meteorology, 149, 1105-1117, 10.1016/j.agrformet.2009.01.008, 2009.

Stengel, M., Stapelberg, S., Sus, O., Schlundt, C., Poulsen, C., Thomas, G., Christensen, M., Carbajal Henken, C., Preusker, R., Fischer, J., Devasthale, A., Willén, U., Karlsson, K.-G., McGarragh, G. R., Proud, S., Povey, A. C., Grainger, R. G., Meirink, J. F., Feofilov, A., Bennartz, R., Bojanowski, J. S., and Hollmann, R.: Cloud property datasets retrieved from AVHRR, MODIS, AATSR and MERIS in the framework of the Cloud_cci project, Earth Syst Sci Data, 9, 881-904, 10.5194/essd-9-881-2017, 2017.

Stubenrauch, C. J., Rossow, W. B., Kinne, S., Ackerman, S., Cesana, G., Chepfer, H., Di Girolamo, L., Getzewich, B., Guignard, A., Heidinger, A., Maddux, B. C., Menzel, W. P., Minnis, P., Pearl, C., Platnick, S., Poulsen, C., Riedi, J., Sun-Mack, S., Walther, A., Winker, D., Zeng, S., and Zhao, G.: Assessment of Global Cloud Datasets from Satellites: Project and Database Initiated by the GEWEX Radiation Panel, Bulletin of the American Meteorological Society, 94, 1031-1049, 10.1175/bams-d-12-00117.1, 2013.

Sun, B. M., Free, M., Yoo, H. L., Foster, M. J., Heidinger, A., and Karlsson, K. G.: Variability and Trends in U.S. Cloud Cover: ISCCP, PATMOS-x, and CLARA-A1 Compared to Homogeneity-Adjusted Weather Observations, Journal of Climate, 28, 4373-4389, 10.1175/jcli-d-14-00805.1, 2015.

Tang, Q., Bo, Y., and Zhu, Y.: Spatiotemporal fusion of multiple-satellite aerosol optical depth (AOD) products using Bayesian maximum entropy method, Journal of Geophysical Research: Atmospheres, 121, 4034-4048, 10.1002/2015jd024571, 2016.

Tiedtke, M.: Representation of Clouds in Large-Scale Models, Monthly Weather Review, 121, 3040-3061, Doi 10.1175/1520-0493(1993)121<3040:Rocils>2.0.Co;2, 1993.

Toll, V., Christensen, M., Quaas, J., and Bellouin, N.: Weak average liquid-cloud-water response to anthropogenic aerosols, Nature, 572, 51-55, 10.1038/s41586-019-1423-9, 2019.

Trepte, Q. Z., Bedka, K. M., Chee, T. L., Minnis, P., Sun-Mack, S., Yost, C. R., Chen, Y., Jin, Z., Hong, G., Chang, F.-L., and Smith, W. L.: Global Cloud Detection for CERES Edition 4 Using Terra and Aqua MODIS Data, IEEE Transactions on Geoscience and Remote Sensing, 57, 9410-9449, 10.1109/tgrs.2019.2926620, 2019.

Tzallas, V., Hatzianastassiou, N., Benas, N., Meirink, J. F., Matsoukas, C., Stackhouse, P., and Vardavas, I.: Evaluation of CLARA-A2 and ISCCP-H Cloud Cover Climate Data Records over Europe with ECA&D Ground-Based Measurements, Remote Sens-Basel, 11, 10.3390/rs11020212, 2019.

Van Tricht, K., Lhermitte, S., Lenaerts, J. T. M., Gorodetskaya, I. V., L'Ecuyer, T. S., Noel, B., van den Broeke, M. R., Turner, D. D., and van Lipzig, N. P. M.: Clouds enhance Greenland ice sheet meltwater runoff, Nature Communications, 7, ARTN 10266

10.1038/ncomms10266, 2016.

Vaughan, M., Young, S., Winker, D., Powell, K., Omar, A., Liu, Z. Y., Hu, Y. X., and Hostetler, C.: Fully automated analysis of space-based lidar data: an overview of the CALIPSO retrieval algorithms and data products, Bba Lib, 5575, 16-30, 10.1117/12.572024, 2004.

Vaughan, M. A., Powell, K. A., Kuehn, R. E., Young, S. A., Winker, D. M., Hostetler, C. A., Hunt, W. H., Liu, Z. Y., McGill, M. J., and Getzewich, B. J.: Fully Automated Detection of Cloud and Aerosol Layers in the CALIPSO Lidar Measurements, Journal of Atmospheric and Oceanic Technology, 26, 2034-2050, 10.1175/2009jtecha1228.1, 2009.

Vignesh, P. P., Jiang, J. H., Kishore, P., Su, H., Smay, T., Brighton, N., and Velicogna, I.: Assessment of CMIP6 Cloud Fraction and Comparison with Satellite Observations, Earth and Space Science, 7, 10.1029/2019ea000975, 2020.

Walsh, J. E., Chapman, W. L., and Portis, D. H.: Arctic Cloud Fraction and Radiative Fluxes in Atmospheric Reanalyses, Journal of Climate, 22, 2316-2334, 10.1175/2008jcli2213.1, 2009.

Wang, D., Bi, S., Wang, B., and Yan, J.: Satellite cloud image fusion based on regional feature with nonsubsampled contourlet transform, Journal of Computer Applications, 32, 2585-2587, 2012.

Winker, D. M., Hunt, W. H., and McGill, M. J.: Initial performance assessment of CALIOP, Geophysical Research Letters, 34, Artn L19803

10.1029/2007gl030135, 2007.

Winker, D. M., Vaughan, M. A., Omar, A., Hu, Y. X., Powell, K. A., Liu, Z. Y., Hunt, W. H., and Young, S. A.: Overview of the CALIPSO Mission and CALIOP Data Processing Algorithms, Journal of Atmospheric and Oceanic Technology, 26, 2310-2323, 10.1175/2009jtecha1281.1, 2009.

Woodruff, S. D., Diaz, H. F., Worley, S. J., Reynolds, R. W., and Lubker, S. J.: Early ship observational data and ICOADS, Climatic Change, 73, 169-194, 10.1007/s10584-005-3456-3, 2005.

Wu, W., Liu, Y. G., Jensen, M. P., Toto, T., Foster, M. J., and Long, C. N.: A comparison of multiscale variations of decade-long cloud fractions from six different platforms over the Southern Great Plains in the United States, J Geophys Res-Atmos, 119, 3438-3459, 10.1002/2013jd019813, 2014.

Xia, X., Zhao, B., Zhang, T., Wang, L., Gu, Y., Liou, K.-N., Mao, F., Liu, B., Bo, Y., Huang, Y., Dong, J., Gong, W., and Zhu, Z.: Satellite-Derived Aerosol Optical Depth Fusion Combining Active and Passive Remote Sensing Based on Bayesian Maximum Entropy, IEEE Transactions on Geoscience and Remote Sensing, 60, 1-13, 10.1109/tgrs.2021.3051799, 2022.

Xie, S. C., McCoy, R. B., Klein, S. A., Cederwall, R. T., Wiscombe, W. J., Clothiaux, E. E., Gaustad, K. L., Golaz, J. C., Hall, S. D., Jensen, M. P., Johnson, K. L., Lin, Y. L., Long, C. N., Mather, J. H., McCord, R. A., McFarlane, S. A., Palanisamy, G., Shi, Y., and Turner, D. D. D.: ARM CLIMATE MODELING BEST ESTIMATE DATA A New Data Product for Climate Studies, Bulletin of the American Meteorological Society, 91, 13-+, 10.1175/2009bams2891.1, 2010.

Xu, S. and Cheng, J.: A new land surface temperature fusion strategy based on cumulative distribution function matching and multiresolution Kalman filtering, Remote Sensing of Environment, 254, 10.1016/j.rse.2020.112256, 2021.

Xu, S., Cheng, J., and Zhang, Q.: Reconstructing All-Weather Land Surface Temperature Using the Bayesian Maximum Entropy Method Over the Tibetan Plateau and Heihe River Basin, IEEE Journal of Selected Topics in Applied Earth Observations and Remote Sensing, 12, 3307-3316, 10.1109/jstars.2019.2921924, 2019.

Yang, J. and Hu, M.: Filling the missing data gaps of daily MODIS AOD using spatiotemporal interpolation, Science of the Total Environment, 633, 677-683, 10.1016/j.scitotenv.2018.03.202, 2018.

Yeo, H., Kim, M.-H., Son, S.-W., Jeong, J.-H., Yoon, J.-H., Kim, B.-M., and Kim, S.-W.: Arctic cloud properties and associated radiative effects in the three newer reanalysis datasets (ERA5, MERRA-2, JRA-55): Discrepancies and possible causes, Atmospheric Research, 270, 10.1016/j.atmosres.2022.106080, 2022.

Young, A. H., Knapp, K. R., Inamdar, A., Hankins, W., and Rossow, W. B.: The International Satellite Cloud Climatology Project H-Series climate data record product, Earth Syst Sci Data, 10, 583-593, 10.5194/essd-10-583-2018, 2018.

Yu, H.-L. and Wang, C.-H.: Retrospective prediction of intraurban spatiotemporal distribution of PM2.5 in Taipei, Atmospheric Environment, 44, 3053-3065, 10.1016/j.atmosenv.2010.04.030, 2010.

Zhang, C.-J., Chen, Y., Duanmu, C., and Feng, H.-J.: Multi-channel satellite cloud image fusion in the tetrolet transform domain, Int J Remote Sens, 35, 8138-8168, 10.1080/01431161.2014.980918, 2014.

Zhang, Q., Cheng, J., and Liang, S.: Deriving high-quality surface emissivity spectra from atmospheric infrared sounder data using cumulative distribution function matching and principal component analysis regression, Remote Sensing of Environment, 211, 388-399, 10.1016/j.rse.2018.04.033, 2018.

Zhu, X., Chen, J., Gao, F., Chen, X., and Masek, J. G.: An enhanced spatial and temporal adaptive reflectance fusion model for complex heterogeneous regions, Remote Sensing of Environment, 114, 2610-2623, 10.1016/j.rse.2010.05.032, 2010.