# Peer review of "A monthly 1-degree resolution dataset of daytime cloud fraction over the Arctic during 2000–2020 based on multiple satellite products"

_Earth System Science Data, 2023_

## Author Comment (AC1)

Dear Reviewer,
We gratefully thank you for your time spent making constructive remarks and suggestion, which helped us improve the quality of the manuscript. Each suggested revision and comment, brought forward by the reviewer was carefully considered and responded. Below the comments of the reviewer are responded point-by-point and the revisions are indicated.

**Overview:**

This manuscript describes studies to reduce satellite retrieved cloud fraction inconsistencies across different products over the Artic region. The inconsistencies can be attributed to differences in sensors, retrieval algorithms, orbital drifts, etc. The authors apply cumulative distribution function (CDF) matching and the Bayesian maximum entropy (BME) method to produce a synthetic monthly 1°×1° cloud fraction fusion dataset in the Arctic during 2000–2020, by utilizing CALIPSO-GEWEX and ground observations as truth data. It is known that there are large uncertainties in cloud fractions derived from passive satellite observations in the Arctic region. The fusion product from this study provides high quality data for the scientific community to use and makes an important contribution. The manuscript is organized and well written. I recommend to accept this manuscript subject to minor but necessary revisions.

**Response:** Thank you for your positive and constructive comments on our manuscript, which given us more confidence in our current and future researches. And we also appreciate your doubts, which helped us think more deeply about the value of this study and improve the quality of the manuscript. We carefully responded your questions about the time period of ICOADS, and we added the uncertainty estimates about the fusion cloud fraction, as well as other questions. We take a revision about the original manuscript according to your comments. Please see the point-by-point responses for details.

**General comment:**

1. The authors only studied sunlit months "because of the darkness of the Arctic winter". However, all the passive sensors and CALIPSO-GEWEX have cloud fraction data at nighttime. The manuscript mentioned that CRU TS data are from sunlit hours, but it seems the ocean data ICOADS are not limited. I would like to see some discussion on the availability and quality of ICOADS data, particularly at nighttime, and if possible to use ICOADS alone as ground truth to derive fusion data for the other months over the ocean.

**Response:** Thank you for comment. Our study had the objective of producing precise measurements of Arctic daytime cloud fraction. Consequently, we exclusively utilized cloud fraction data labeled as "daytime" from several satellite datasets. We agree with your perspective that "CRU TS data are from sunlit hours, but it seems the ocean data ICOADS are not limited". The ICOADS dataset contains cloud fraction observed in daytime and nighttime. In order to use daytime data as much as possible, we only applied the cloud fraction data with a "fraction of observations in daylight" exceeding

0.8. The data with the label of "fraction of observations in daylight" is also contained in the ICOADS dataset (https://psl.noaa.gov/data/gridded/data.coads.1deg.html). For the sake of clarity, we included a detailed explanation about this methodology in our manuscript. (**Lines197-199, Page5; Lines248-252, Page7**)

2. One important but sometime missing aspect of satellite products are the uncertainties of the retrieved variables. Are there any uncertainty estimates of the fusion cloud fraction from the CDF-BME methods? If yes, can the authors show some plots?

**Response:** Thank you for pointing this out. We have added the uncertainty estimates of the fusion cloud fraction in our manuscript. We use the error standard deviation which generated by the BME fusion process, please see section 5.3 (**Lines 743-756, Pages 29-30**). The contents are as follows:

*To assess the fusion algorithm's reliability, we calculated the standard deviation of error within each grid value in the fusion process. Specifically, we determined the standard deviation of the predicted posterior probability density function on each grid point. Our findings demonstrate that, with the exception of the northern region of Greenland and part of the margin error, the standard deviation of error in other areas was within 3% (FIG. 4-16). We attribute these discrepancies primarily to the underestimation of ground and satellite observations by satellite data, particularly ISCCP-H data, by around 10-30% in the central zone of Greenland. Moreover, the CF of ISCCP-H was significantly overestimated beyond the Greenland margin. Such significant inconsistencies can adversely affect the fusion results. Moreover, because the CF of satellite data, particularly satellite data based on AVHRR, was significantly lower than that of ground observation data and active sensor data in April, and a significant difference existed between different datasets, the standard deviation of error after fusion marginally increased in April, with some areas at approximately 4%.*

[Figure]

*Figure 15. The mean error standard deviation of the fusion results*

3. Four MODIS-sensor based products are used here (MYD35, MOD35, CERES Aqua, CERES Terra). Are the authors aware of the MODIS-VIIRS continuity product? The CLDPROP MODIS data from the continuity product can certainly add values to this study.

**Response:** We thank you for reminding us this important point. The NASA Aqua MODIS and Suomi National Polar-Orbiting Partnership (SNPP) Visible Infrared

Imaging Radiometer Suite (VIIRS) climate data record continuity cloud properties products (CLDPROP) were publicly released in April 2019 with an update later that year (Version 1.1). These cloud products, having heritage with the NASA Moderate-resolution Imaging Spectroradiometer (MODIS) MOD06 cloud optical properties product and the NOAA GOES-R Algorithm Working Group (AWG) Cloud Height Algorithm (ACHA), represent an effort to bridge the multispectral imager records of NASA's Earth Observing System (EOS) and NOAA's current generation of operational weather satellites to achieve a continuous, multi-decadal climate data record for clouds that can extend well into the 2030s. This product ensures continuity of approach through a common algorithm that is applicable to both MODIS and VIIRS data by leveraging only those spectral channels that are common to both instruments.

The L3 monthly Cloud Properties product is derived by aggregating the Aqua/MODIS D3 Cloud Properties product (CLDPROP_D3_MODIS_Aqua), which is a global gridded dataset that is produced daily. This monthly product contains 128 science data sets (SDS/parameters), as well as the daytime cloud fraction. Like CLDPROP_M3, the MYD08 Level-3 product is a 1° equal angle aggregation of the Level-2 pixel-level MODIS retrievals, but includes all MODIS Atmosphere Discipline datasets in addition to the cloud datasets. Though the codes and production facilities that produce the MYD08_M3 and CLDPROP_M3 aggregations are different, tests have shown that the CLDPROP_M3 results are indeed consistent with MYD08 when ingesting the MYD06 Level-2 products. We believe that the CLDPROP MODIS data from the continuity product can help us to extend the fusion cloud fraction over a longer period of time.

We intend to employ the CLDPROP MODIS data and other relevant datasets in our forthcoming research to enhance the fusion outcomes.

**Other specific points:**

1. Ln 278: 90% percentile -> 90 percentile

**Response:** Thank you for your careful check, we have corrected it (**Line 327, Page 9**).

2. Ln 289: add "of" after "the time series"

**Response:** We appreciate your attention to detail and we have now included the designated word (**Lines 339, Page 10**).

3. Ln 306-308, 418-420: It's unclear to me if the authors apply relationship derived from latitudes less than 82.5N to higher latitude beyond calipso coverage. And where are the bias to CPCF relationship plots? Figure 5 only shows bias and CF against SIC. Does Figure 5 indicate CF is stable as SIC increases and starts to decrease when SIC is very high? Why is that?

**Response:** We are sorry for the confusion brought to you. In this manuscript we implemented correction for the passive sensor data in regions with latitudes that exceed 82.5°N. The aforementioned amendment is rooted in a strong correlation identified between the bias present in the passive sensor data following and prior to CDF matching, and the cumulative percentage of CF (CPCF) and sea ice concentration (SIC) detected in sea ice regions situated within latitudes below 82.5°N. The CPCF means the average CF over an interval of SIC, and the interval, which in this manuscript is 1%. In Figure

5 the mean of bias increased with the SIC, the CPCF appeared to decrease with increasing SIC, a negative correlation between CPCF and bias was also evident. The depicted fitting curves in Figure 5 have considered the influence of CPCF. (**Lines 483-484, Page 17; Lines 489-493, Page 18**)

In regards to the last question, figure 5 does indicate that CF is stable (it's actually slightly decreased) as SIC increases and starts to decrease when SIC is very high. It was observed that the passive sensor's ability to detect clouds was impacted by higher levels of sea ice concentration. As a result, the sensor tended to underestimate CF, especially near the center of the Arctic Ocean. Finally, the relationship between the CPCF and SIC displayed a tendency to decrease as the sea ice concentration increased.

4. Ln 523-525: "original satellite data", should they be fused data?

**Response:** We apologize for our confusing words and we have corrected it to "fused data". We have conducted a comprehensive review of the complete manuscript to ascertain that any identical errors have been rectified. (**Line 594, Page 23; Line 595, Page 24**)

**Thank you again for your constructive comments and suggestions on our manuscript. There is no doubt that these comments are valuable and very helpful for revising and improving our manuscript. We hope you will find our revised manuscript acceptable for publication.**

---

## Author Comment (AC2)

Dear Reviewer,
We gratefully thank you for your positive and constructive comments on our manuscript, which helped us improve the quality of the manuscript. We carefully considered and responded each suggested revision and comment, and we hope this manuscript can be further improved. We have provided a point-by-point response to the comments raised by the reviewers below, and we have indicated the revisions made in the manuscript.

**Overview and General recommendation:**

1. This manuscript describes methods used to create a fused cloud fraction (CF) dataset for the Arctic region using passive and active satellite products as well as ground observations. The authors additionally evaluate the fused cloud product and corrected individual satellite cloud products against additional datasets. The methods described are as follows: After initial data quality control, the authors match the cumulative probability distribution of the passive satellite products to more robust active satellite products. Following adjustment to the original data, the authors derive a spatio-temporal covariance function from the observations that is then used in a Bayesian Maximum Entropy (BME) method to produce a filled CF record. Following the presentation of these methods, the authors demonstrate that these observations have generally improved biases in passive satellite records of Arctic clouds.

I find this manuscript to be generally well-written and organized. The content is quite thorough and appropriate for the audience of ESSD. The need for an Arctic cloud fraction dataset that combines the advantages of different observational products is well-motivated as well. While the manuscript is thorough, some concepts lack appropriate description or are presented without introduction. Additionally, I found some figures to be confusingly introduced, described, and labelled. These aspects of the presentation of the material should be improved before publication. Finally, the authors should discuss the uncertainties in their data product in the discussion. Specifically, is uncertainty quantified by the methods employed? If not, why not and what are the major sources of uncertainty? I have thoroughly documented my comments below.

**Response:** We truly appreciate your time and effort in providing us with a positive review. Thank you very much for your valuable feedback on our manuscript, which has significantly improved the presentation of our manuscript. We have carefully considered all of your suggestions and have made the necessary revisions to our manuscript in accordance with your recommendations. For some inappropriate concepts, we re-described them and added the introductions. For the confusing figures which you proposed in the specific comments, we revised them according to your suggestion. The discussion about the uncertainties in the fusion CF was added in the discussion section. We have also explained and rewrote the comparison with model data. The manuscript has also been double-checked, and the typos and grammar errors we found have been corrected. In the following section, we summarize our responses to each comment from the reviewers. We believe that our responses have well addressed all concerns from the reviewers.

Please see the point-by-point responses for details.

2. My only other comment concerns the use of a CMIP6 model as an independent testing dataset.

(1) Global climate models in general are not a good independent testing dataset for the presented CF product. Clouds are highly parameterized in global model output (even at 25 km resolution, convection and other key processes cannot be resolved). In the Arctic, model performance regarding cloud fields is especially poor, in part due to the complex nature of the mixed-phase clouds present for much of the year (https://www.nature.com/articles/ngeo1332). Comparing an observational cloud field against a global climate model to validate the observational data does not seem appropriate. In summary, this reviewer does not think that global models should be used as a verification dataset in this work. Use of the ERA5 reanalysis is appropriate, but not that of global models.

**Response:** Thank you for raising this matter. We must apologize for this error expression. We are real intended to show that the fusion data is more consistent with the model data and the reanalysis data, which shows that the fusion products have a helpful role in reducing the uncertainty of cloud fraction. As a result, we have deemed it is not appropriate to remove the comparison with the CMIP6 model in the text.

(2) The language of the paper states that CMIP6 is used for independent testing of the CF product, when only a single CMIP6 model is used. This language is confusing and perhaps misleading. If the model comparison is kept (which I discourage), the authors should specify that a single CMIP6-generation model is used.

**Response:** We are sorry for this confusing and misleading. In the original manuscript we use one of the CMIP6-generation model, the MRI-AGCM3-2-S climate model to test the fusion CF. And in the revised manuscript, we corrected it as your suggestion. Thank you for your valuable suggestion.

3. Overall, this reviewer finds the submitted manuscript to be well-motivated and structured. Aside from my comment regarding the model comparison, this paper merits publication when the presentation of concepts and results has been improved. I recommend minor revisions.

**Response:** We sincerely appreciate the positive feedback and extend our gratitude to the esteemed reviewer for their insightful comments and recommendations, which have contributed immensely to enhancing the caliber of our manuscript. We have carefully reviewed the reviewer's comments and have made the necessary revisions to ensure the manuscript meets the highest standards.

**Specific comments:**

1. 21-24: Differences between satellite products are consistently referred to as biases relative to the "true" values derived from active sensors. While passive sensors are

not as well-suited for cloud retrieval, some differences arise purely from differences in instrumentation and the definition of a cloud field. Differences in instrumentation impose these different cloud definitions, and have inspired "satellite simulator" software designed to take definitional differences into account (https://journals.ametsoc.org/view/journals/bams/92/8/2011bams2856_1.xml). The approach described here brings different instruments to a common standard, which addresses both definitional differences and biases. I recommend noting this nuance in the introduction (though not the abstract).

**Response:** We agree with the reviewer's point and have added the necessary description to clarify the issues raised. In **Lines 55-58, Page 2**, we proposed that the variances in CF definitions and system differences commonly exist among different sources of data. However, the fused product can reduce the uncertainties caused by definition and system differences. In **Lines 123-125, Page 3**, we also indicated that the differences in instrumentation impose these different cloud definitions can further larged the biases between the passive sensor data and the active sensor data.

2. 33-35: I find this sentence confusing. It seems that the biggest outliers are the most reduced by this method?

**Response:** We are so sorry for this confusing. We have revised the sentence to improve its clarity. The use of reanalysis data for verification is intended to demonstrate if fusion data can significantly improve the consistency between satellite observation data and reanalysis data. The sentence has been replaced as "*The results of the comparison with the ERA5 and the MRI-AGCM3-2-S climate model suggest an obvious improvement in the conformity between the satellite-observed CF and the reanalysis data via fusion. This serves as a promising indication that the fused CF results hold the potential to deliver reliable satellite observations for modeling and reanalysis data.*" (**Lines 34-37, Page 1**)

Is it effectively removing the worst errors in the passive obs?
The question has been answered through comparisons with ground observations and active sensor data. The worst errors of the passive sensor data occurred in Greenland and Sea Ice regions, and the inconsistencies of Arctic CF between passive sensor products and the reference data (the ground observations and the CALIPSO data) were reduced by about 10–20% after fusing, with particularly noticeable improvements in the vicinity of Greenland. (**Lines 26-34, Page 1**)

3. 1-52: The fused product is more definitionally consistent as well as accurate than other datasets.

**Response:** We appreciate your kind suggestions and we have added this sentence in our manuscript to make its research value more prominent (**Lines 55-58, Page 2**):
*However, variances in CF definitions and system differences commonly exist among different sources of data. As a solution, the fused product provides a higher level of definition consistency and accuracy in comparison to alternative datasets.*

4.  71-72: Sentence starting with "However," is confusing and vague.

**Response:** Thank you for pointing this out, we have revised this sentence (**Lines 77-78, Page 2**).
"*It should be noted that the differences in CF may have a more obvious impact on the surface radiation budget in high-latitude polar regions.*"

5.  127-130: Sentence transition is confusing.

**Response:** We apologize for that, and we have rewritten this sentence as "*Given that passive sensor CFs exhibit seasonal fluctuations similar to those of active sensor data (peaking in September and minimizing in April in the Arctic), an approach based on cumulative distribution function (CDF) matching using time series data may be able to improve both the accuracy and efficiency of CF detection.*" (**Lines 136-139, Page 4**)

6.  183-188: What about the redundancy of using products developed from the same observations? Even though processing algorithms may differ, products based off of the same observations are not independent. Is this accounted for by the methods? Does duplicate data make predictions over-confident?

**Response:** We appreciate the feedback provided by the reviewer. Bayesian maximum entropy (BME) is a knowledge-centered approach that can enhance the accuracy of inference by treating observation data with uncertainty as soft data. Research has demonstrated that higher quality soft data, i.e., those with relatively lower uncertainty can lead to more accurate BME predictions. Furthermore, as the amount of reliable information increases, the accuracy of the prediction also rises (He et al., 2017; D'Or et al., 2001). In our study, we have utilized the cumulative distribution function (CDF) matching method to improve the accuracy of each type of passive sensor data. Therefore, we believe that these processed observational data can provide valuable reference information for predicting results. Therefore, appropriately identifying and handling "duplicate data" can potentially enhance the precision and effectiveness of prediction models.

References:
D'Or D, Bogaert P, Christakos G. Application of the BME approach to soil texture mapping. Stoch Environ Res Risk Assess 15:87–100.
He, J. and Kolovos, A.: Bayesian maximum entropy approach and its applications: a review, Stochastic Environmental Research and Risk Assessment, 32, 859-877, 10.1007/s00477-017-1419-7, 2017.

7.  208-209: What does "noticeable" mean in this context. This sentence is confusing.

**Response:** We should apologize for this confusing description, we have revised this sentence as "*Although some differences exist between Terra and Aqua, the consistency between these two satellites cannot be ignored.*" (**Lines 219-220, Page 6**)

8.  239-245: Please discuss the limitations of reanalysis of cloud fields in the Arctic. Like global models, reanalysis often struggles to capture realistic cloud fields, but due to a lack of consistent observations.

**Response:** We appreciate the constructive criticism you provided. We have rewritten the section of 2.3 Reanalysis Data and Model Data and have taken your suggestions into account in the revised version of the manuscript. (**Lines 253-278, Pages 7-8**)

9.  246-251: See previous comments regarding the use of global models as testing data for this dataset. Separately, under what simulation was global model data obtained? Has this model been evaluated in its ability to capture clouds in the Arctic?

**Response:** Thank you for your comment. As per your suggestion, we have rewritten the comparison with model data from the manuscript. In our original manuscript, we used model data obtained from the MRI-AGCM3-2-S climate model (**Lines 279-295, Page 8**). This model has been evaluated for its ability to capture clouds on a global scale in previous research (Sugi, 2012; Mizuta et al., 2012; Kusunoki, 2018). However, we have not yet come across a study that specifically evaluates the model's cloud detection capabilities in the Arctic.

**References:**

Sugi, M.: Changes in Earth's Energy Flows and Clouds in 228-Year Simulation with a High-Resolution AGCM, Surveys in Geophysics, 33, 427-443, 10.1007/s10712-012-9183-1, 2012.

Mizuta, R., Yoshimura, H., Murakami, H., Matsueda, M., Endo, H., Ose, T., Kamiguchi, K., Hosaka, M., Sugi, M., Yukimoto, S., Kusunoki, S., and Kitoh, A.: Climate Simulations Using MRI-AGCM3.2 with 20-km Grid, Journal of the Meteorological Society of Japan. Ser. II, 90A, 233-258, 10.2151/jmsj.2012-A12, 2012.

Kusunoki, S. Is the global atmospheric model MRI-AGCM3.2 better than the CMIP5 atmospheric models in simulating precipitation over East Asia?. Clim Dyn 51, 4489–4510 (2018). https://doi.org/10.1007/s00382-016-3335-9

10. 258-262: Please explain the third and fourth steps in plain language, the meaning of these sentences is unclear. Additionally, please define heterogenetic and isotropous.

**Response:** We would like to thank the reviewer for your thoughtful comments and suggestions, and we have made the necessary revisions to ensure the manuscript meets the highest standards (**Lines 304-309, Pages 8-9**). The explanations of the third and fourth steps are as follows:

**For the third step:** BME theory is founded on the space-time random field hypothesis (S/TRF), which provides a robust theoretical framework for studying natural phenomena that evolve in a space and/or space-time continuum. When S/TRF is spatially and temporally stationary, the covariance function is affected only by the relative distance between any two positions. This implies that all the variables used in the process are homogeneous and isotropic. However, natural processes, such as cloud fraction distribution, typically exhibit anisotropy, varying both in time and space. Nonetheless, the natural spatio-temporal variation process can be broken down into a heterogeneous global spatio-temporal trend component and a spatio-temporal isotropic component. To comply with BME's second-order stationarity prerequisite that assumes

constancy of mean and variance, it is imperative to remove the global spatio-temporal trend before estimating the spatio-temporal autocorrelation structure of the data. In this study, the data obtained from various passive sensors was combined, and subsequently, the average CF value within a 5°(longitude) × 5°(latitude) × 3 (months) spatio-temporal filtering window was used to determine the global space-time trend component for the center location. The residuals were obtained through the deduction of the heterogenetic global trend component of CF from the original CF data for each grid. It was posited that the residuals maintained spatiotemporal stationary characteristics and utilized for spatiotemporal fusing.

**For the fourth step:** In spatiotemporal geostatistics, the covariance function is a measure of the spatial and temporal dependence of the data, which tend to lessen as distance or time increases. Similarly, the spatio-temporal variation of CF can be represented by a spatio-temporal covariance function. Our study presents the spatiotemporal covariance model of CF data first, followed by an update of the prior probability density function incorporating this knowledge as a fundamental component towards deriving a posterior probability density function.

11. 273: Please specify what you mean by standard deviations here. Standard deviations across datasets, within datasets? The language should be specified to make this sentence clear.

**Response:** Thank you for your comment, we have specified the language: *For satellite datasets, statistics always have the Scientific Data Set (SDS) name suffix "_Standard_Deviation" and which are computed by calculating an unweighted standard deviation of all pixels or samples within a given 1° grid cell.*" (**Lines 320-322, Page 9**)

12. 288: "imposes the value range", the meaning of this phrase is unclear.

**Response:** We have revised this as: "*Several studies have proved that the process of adjusting this distribution does not change the variation of original satellite-based products, but rather aligns the value range with that of the reference data.*" (**Lines 335-338, Page 10**)

13. 307-308: How does the uncertainty of the new dataset differ between regions with the true CDF matching and with the correction based on sea ice concentration?

**Response:** We appreciate the thoroughness of the reviewer's feedback, and we have discussed this question in section 4.1. From Figure 6, the corrected CFs based on the sea ice concentration (SIC) and the cumulative percentage of CF (CPCF) have consistency with the CFs corrected by the CDF matching, with $R^2$ over 0.75, RMSE less than 3.6, and bias less than 0.5. And from Figure 7, the frequency of the standard deviation (STD) of multiple satellite CFs reduced about 3.02% for regions with latitude less than 82.5°N and reduced about 4.51% for regions with latitude over 82.5°N. However, the distribution of STD frequency in regions over 82.5°N and in the entire sea ice area seemed similar, with the difference of reduction less than 0.05%. (**Lines 495-523, Pages 18-23**)

14. 328-329: Figure 2, panel b. I don't understand the units here or what this panel is showing. I expected the x-axis to be the trend that was calculated and removed.

**Response:** We thank the reviewer for their constructive criticism and helpful suggestions. This figure is (a) the statistical descriptions of original satellite CF, (b) global spatiotemporal trend, and (c) spatiotemporally isotropous component, for the entire Arctic area in 2010 (the distributions were similar in other years). The aforementioned figure was utilized as evidence to demonstrate the characteristic of the residual component as being approximately normally distributed. This aligns with the prerequisite for accurately modeling the structure of spatiotemporal autocovariance. The x-axis represents the variable of interest, in this study it is the CF.

15. 344: Covariance parameters are modeled separately by year. What about my geographic region in the Arctic? Is the Arctic domain small enough to justify a single fit to the spatial parameters?

**Response:** Thank you for pointing out this problem in our manuscript. According to the revised content, we have added the discussion about the spatial range of the covariance parameters. Based on the modelled results, the model has a spatial range of 2°, a temporal range of 3 months, and a partial sill variance of 0.85 for local scale CF (the first nested covariance model). And for the large range CF the model has a spatial range of 30°, a temporal range of 6 months, and a partial sill variance of 0.15 (the second nested covariance model). (**Lines 398-402, Pages 11-12**)

16. 346: Please define what you mean by soft data.

**Response:** We have defined it: *BME treated the informative content with uncertainty from different sources as soft data (He and Kolovos, 2017). For example, the observed data that accompanied by obvious sources of uncertainty such as inaccuracy in measuring devices, modeling uncertainties, and human error. In this study, the CF data of passive sensor products are viewed as soft data.* (**Lines 405-408, Page 12**)

17. 384: I think this equation has an error. The left side should be the conditional probability of x_k given the soft data from the observations. I.e. I expected f(a|b) = f(a,b) / f(b)

**Response:** Thank you for your careful check, we have corrected it (**Line 447, Page 14**).

$$f\left(x_k \mid x_{soft,1}, x_{soft,2}...x_{soft,n}\right) = \frac{f\left(x_{soft,1}, x_{soft,2}...x_{soft,n}, x_k\right)}{f\left(x_{soft,1}, x_{soft,2}...x_{soft,n}\right)}, \tag{7}$$

18. 386: Is the equation on this line also incorrect?

**Response:** Yes, thank your careful work again, and we have revised it as :

$f(x_{soft,1}, x_{soft,2}...x_{soft,n}, x_k)$ (**Line 449, Page 14**)

19. 419-420: This is not clear to me after viewing Figure 5.

**Response:** We are so sorry to have caused you such trouble, Figure 5 shows the

relationship among the bias present in the passive sensor data following and prior to CDF matching, the cumulative percentage of CF (CPCF) and sea ice concentration (SIC) detected in sea ice regions situated within latitudes below 82.5°N. The CPCF means the average CF over an interval of SIC, and the interval, which in this manuscript is 1%. We implemented correction for the passive sensor data in regions with latitudes that exceed 82.5°N based on this relationship. We have added the information in the manuscript (**Lines 481-502, Page 17-18**).

20. 421-422: Figure 5. I do not understand what this figure is trying to show. I see the bias and CF as a function of SIC, but I do not see how the bias has changed after the CDF matching step.

**Response:** Thank you for the comment. Figure 5 shows the relationship among the bias present in the passive sensor data following and prior to CDF matching, the CPCF and the SIC detected in sea ice regions situated within latitudes below 82.5°N (Comment 19). The fitted results are shown by blue lines in every panel. The results indicated that the mean of bias increased with the SIC. Moreover, the CPCF appeared to decrease with increasing SIC, a negative correlation between CPCF and bias was also evident (**Lines 489-502, Page 18**).

21. 425-426: This sentence is confusingly worded.

**Response:** We have rewritten these sentences: *By virtue of this association, SIC and CPCF are modeled as dependent variables of the bias. Due to the predominant presence of sea ice over the domain located above 82.5N, we employ this functional association to remediate CF inaccuracies in the region, called C-SIC Corrected CF.* (**Lines 489-491, Page 18**)

22. 426-427: Figure 6 compares the "corrected CF" with CALIPSO data, but the introduction of this figure indicates that the data is for latitudes higher than 82.5N, where CALIPSO does not sample. The way that Figure 6 is presented is very confusing. You can't evaluate the corrected CF product north of 82.5N using CALIPSO, right?

**Response:** We are so sorry for this confusing phenomena. In this manuscript we implemented correction for the passive sensor data in regions with latitudes that exceed 82.5°N. The aforementioned amendment is rooted in a strong correlation identified between the bias present in the passive sensor data before and after CDF matching, and the cumulative percentage of CF (CPCF) and sea ice concentration (SIC) detected in sea ice regions situated within latitudes below 82.5°N. By virtue of this association, SIC and CPCF are modeled as dependent variables of the bias. Due to the predominant presence of sea ice over the domain located above 82.5°N, we employ this functional association to remediate CF inaccuracies in the region, called C-SIC Corrected CF. (**Lines 489-506, Pages 18-20**)

23. 434-437: This label should specifically describe the region over which CF fields are compared. Also, the meaning of the third panel is confusing and should be described more clearly in the figure caption and in the text.

**Response:** Thank you for your helpful suggestion, we have revised it as your comment.

*Figure 6's initial two panels depict a comparison between the CF of active data and passive data before and after correction by C-SIC in sea ice regions below 82.5°N.* (**Lines 491-493, Page 18**). '*The third panel of Figure 6 shows the comparison of C-SIC Corrected CF and the CDF matching CF in sea ice regions with latitude less than 82.5°N.*'(**Lines 496-498, Page 18** ). '*Figure 6. The scatter plots of the cloud fraction (CF) comparison between the passive sensor datasets and the active sensor dataset before (the first panel) and after (the second panel) using the method of CF corrected by the cumulative percentage of CF and SIC (C-SIC). And the scatter plots of the results comparison between C-SIC and cumulative distribution function matching (the third panel).*' (**Lines 503-506, Page 20**)

24. 439-452: This paragraph is generally difficult to read. Figure 7 does a good job presenting the results numerically. The written portion should focus on clear descriptions of what the reader should take away from Figure 7. Lines 442-444 are especially difficult to interpret and required multiple readings.

**Response:** We sincerely apologize for the inconvenience of reading caused by confusing description, and we also thank you very much for your recognition of figure 7. We have revised the manuscript according to your suggestions.

*Figure 7 displays the standard deviation between 1° × 1° passive sensor CF data before and after the application of cumulative distribution function matching (latitude≤82.5°N) and C-SIC correction (latitude >82.5°N). The results obtained from different regions indicate an obvious decrease in the inconsistency between multiple passive sensor data after the correction with the aforementioned methods. In the Holarctic region, multiple passive sensor CFs saw a decrease in mean STD from 9.18% to 5.75%, with more than 50% of the corrected data displaying a standard deviation within 5%. The sea ice region saw the largest reduction rate of the mean STD, approximately 4.5%. This reduction was mainly derived from a STD value range of 10–15%, due to the limited detection capacity of passive sensor data in sea ice areas. Regions with latitude less than 82.5°N saw a decrease in mean STD of only 3.02%. In contrast to the sea ice region, these land regions saw a smaller standard deviation between multiple satellite data. The distribution of STD frequency in regions over 82.5°N and the entire sea ice area appeared similar, indicating that the C-SIC correction method was highly effective in 82.5°N regions.* (**Lines 511-522, Pages 20-21**).

25. 454-455: What correction methods are being shown in the >82.5N region?

**Response:** We have included an appropriate description to enhance the clarity and comprehensibility of the sentence.

*Figure 7 displays the standard deviation between 1° × 1° passive sensor CF data before and after the application of cumulative distribution function matching (latitude≤82.5°N) and C-SIC correction (latitude >82.5°N).* (**Lines 525-526, Page 21**)

26. 485-488: This is an excellent clear summary sentence. The use of the word "significant" here and in other parts of the manuscript indicate that statistical significance testing has been done comparing the initial and final data product, but this

analysis is not presented. I think that the results are appropriately presented, but the word "significant" should not be used to describe them unless formal significance testing is presented.

**Response:** We appreciate your kind suggestions and we have replaced this word by "obvious" accordingly. we have verified and rectified comparable instances of misuse throughout the manuscript.

27. 491-492: Can you re-iterate that the fused dataset only covers a sub-section of the year because it relies on passive sensors? The timeseries in Figure 9 could also be modified to explicitly show the months where the data fusion process is not used.

**Response:** Thank you for your helpful suggestion, we have revised the manuscript and modified the timeseries in Figure 9 according to your suggestions.

*Figure 9 depicts the fluctuation of the mean value on a monthly basis for all data during sunshine periods (April to September) before and after fusion, as demonstrated by the time series.* (**Lines 562-564, Page 22**).

[Figure]

*Figure 9. The area-weighted means of cloud fraction over (a) Holarctic, (b) Land, and (c) Sea for different products in the Arctic from April to September during 2000 to 2020. The time ranges for ISCCP-H and CALIPSO-GEWEX were from 2000 to 2017 and from 2006 to 2016, respectively.*

28. 500-501: Use of the word famous is confusing.

**Response:** We sincerely apologize for the inconvenience of reading caused by verbal error, and we have replaced it by "*Well-known*" (**Line 572, Page 23**). Accordingly, we have corrected the problems and all similar ones throughout the manuscript without altering the paper's original meaning.

29. 516: Qualitative, not qualitatively.

**Response:** We have revised it (**Line 587, Page 23**).

30. 523-525: I do not know what the term "original satellite data" refers to. Doesn't the fused product have the higher $R^2$ and lowest RMSE values?

**Response:** We are so sorry for this expression error. As you said, the fused product has the higher $R^2$ and lowest RMSE values. We have corrected it and checked the full manuscript (**Line 594, Page 23; Line 595, Page 24**).

31. 554-556: Can you comment on why this might be the case? Does this imply biases in the active sensor data? Biases in the ground station data?

**Response:** We thank your comment. There exists a significant systematic deviation between satellite and ground CF, attributable to differences in observation perspectives and CF definitions. Consequently, the divergence between satellite data is comparatively minor when compared to the variation between satellite and ground data.
**Reference:**
Liu, X., He, T., Sun, L., Xiao, X., Liang, S., and Li, S.: Analysis of Daytime Cloud Fraction Spatiotemporal Variation over the Arctic from 2000 to 2019 from Multiple Satellite Products, Journal of Climate, 35, 3995-4023, 10.1175/jcli-d-22-0007.1, 2022.

32. 557: See previous comment on "significant" results.

**Response:** We have revised it (**Line 628, Page 25**).

33. 582-583: I would not report this result if the data is not shown.

**Response:** We thank the reviewer's kind comments and we have removed it.

34. 592-593: Indeed, I don't think you need to include the model output as a testing dataset given how poorly clouds are represented. Is there any reason to think that the model is better than ground observations or reanalysis? What does this comparison add to the conclusions of this paper?

**Response:** Thank you for bringing this to our attention again. We used the comparison with the reanalysis data and the model data to explain that the fused cloud fraction could reduce the uncertainties of the observations, which has the potential to provide reliable constraints on models and reanalysis data. Please see Comment 1 and Comment 2.

35. 605: Please discuss why this is true. Lower cloud fractions over land are better connected by the CDF process? What does this mean about biases over ocean in the central Arctic where active satellite observations are missing?

**Response:** Thank you for pointing out this problem. This can be attributed to the fact that the original satellite data over land regions, notably in Greenland, possess less CFs compared to ground site data. The application of CDF matching correction method led to a noteworthy improvement in addressing this underestimation. Conversely, the satellite data over ocean regions have shown a greater average value compared to marine station observations, attributable to the irregularity in sampling at ground stations. After implementing CDF matching correction, the overestimation for these data reduced correspondingly. It can be inferred that "These improvements were more obvious for CFs over land regions (for underestimated CFs)." The consistency of the two algorithms utilized in regions with (CDF matching method) or without (C-SIC Corrected CF) active satellite observations has thoroughly addressed in Section 4.1

36. 618-619: This sentence is confusing. The correction reduces the number of underestimates, but increases the number of overestimates as a consequence?

**Response:** We must apologize for this confusing description, and we have rewritten it as "*Satellite observation covering open sea areas typically presents a higher CF compared to station observation. Consequently, partial overestimation may persist despite correction by the CDF matching approach.*" (**Lines 691-693, Pages 27-28**).

37. 631-632: Where is this shown?

**Response:** We have added the figure in section Appendix A, Figure A1 (**Lines 817-821, Page 32**). The sentence was revised by: *The findings indicate that any deviations in matching parameters were under 0.05% when the time horizon exceeded 8 years. This demonstrates a level of stability in the correction coefficient when utilizing data for a period exceeding 11 years (Figure A1).* (**Lines 704-707, Page 28**).

38. 632-633: This figure is confusing to me. Is the CF difference the change in CF for each satellite before and after matching?

Is the CF difference the change in the difference between the satellite and ground observations before and after matching?

The figure description is unclear. If it is the second, this is just the change in cloud fraction before and after matching because the observations are unchanged, right?

**Response:** The CF difference is the change in the difference between the satellite and ground observations before and after matching. We are sorry for the confusing writing and we have revised it by "*Figure 14 displays the variation in differences between satellite data and ground observations before and after conducting CDF matching throughout the duration of the study. These differences are calculated by subtracting the deviation between satellite data and ground observations subsequent to CDF matching from that prior to CDF matching.*" (**Lines 707-710, Page 28**).

Just as you motioned, the essence of this difference is the change in CF before and after matching. However, in view of data gaps present in some of the original satellite data, the difference derived in our manuscript does not precisely equivalent to the change in CF before to and after to matching.

39. 652-653: Is this result shown? If not I would just discuss it more generally and not reference results that are not in the manuscript.

**Response:** It is really true as reviewer suggested that the reference results that are not in the manuscript is not suitable to be discussed in detail. We have revised it as your suggestion.

*In this study, we constructed soft data for CF over land, ocean, and GrIS regions every month separately by analyzing the PDF differences for different regions and different months, which realized more consistent results with the ground observations.* (**Lines 737-739, Page 29**).

40. 711-712: This adds valuable physical insight. I would mention this earlier in the

discussion as well.

**Response:** Thank you for your careful reading of our manuscript, we are glad to see your recognition.

**We hereby resubmit the revised manuscript and hope that all corrections are satisfactory. Please feel free to contact us with any questions and we look forward to your decision.**

---

## Author Response (AR2)

Dear Luis Millan,

We gratefully thank you for giving us the opportunity to revise the manuscript again, your expertise and thoughtful suggestions have greatly contributed to the improvement of our work. We carefully reviewed and considered each of your suggestions and recommendations. We are pleased to inform you that we have incorporated your feedback into the revised version of the paper, and the entire manuscript has been thoroughly checked. Below the comments are responded point-by-point and the revisions are indicated.

Public justification (visible to the public if the article is accepted and published):

Please address the following:

1. Since the manuscript is only studying about daytime cloud fraction, I suggest adding "daytime" to the title;

**Response:** Thank you for your suggestion, we have added "daytime" before the words "cloud fraction", the title is "*A monthly 1-degree resolution dataset of daytime cloud fraction over the Arctic during 2000–2020 based on multiple satellite products*" now. **Line 1, Page 1.**

2. Line 748: "underestimation of ground and satellite observations by satellite data, particularly ISCCP-H data". This reads misleading. Please rephrase.

**Response:** We are sorry for the confusion brought to you, we have revised this sentence to "We attribute these discrepancies primarily to the underestimation of satellite observations, particularly the ISCCP-H data, by around 10-30% in the central zone of Greenland." **Lines 747-749, Page 30.**

We have checked the manuscript and made some revisions, all of which are marked up in yellow.

Some slight revisions can be seen in Line 29, Page 1; Line 124, Page 3 and Line 181, Page 5.

At the end paragraph of the discussion of **5.3 The uncertainties of the fusion CF**, we added the discussion about the uncertain of fused CF in the Greenland region, "*It should be noted that our fused data shows an overestimation in the Greenland region. This is mainly because the fusion process prioritizes consistency between the fused data and ground observations. In specific applications, users can make corresponding adjustments based on active sensor data for calibration purposes.*" **Lines 754-757, Page 30.**

We also added two financial support in **Lines 850-851, Page 33**: *Hubei Natural Science Foundation Grant (2021CFA082) and National Key Research and Development Program of China (2020YFA0608704)*

**Thank you again for your constructive comments and suggestions on our manuscript. There is no doubt that these comments are valuable and very helpful for revising and improving our manuscript. We hope you will find our revised manuscript acceptable for publication.**

---

## Author Response (AR3)

Dear Luis Millan,

We greatly thank you for your careful check, we have reviewed it and made the necessary corrections (Page 3 line 124).

And thank you for the time spent by the editors and anonymous reviewers. All the historical comments are valuable and have been very helpful in revising and improving our manuscript. We hope that you will find our revised manuscript acceptable for publication.